# Multi-path Reasoning on a Budget: Towards Theoretically Optimal Hyperparameter-free Adaptive Self-consistency

## Abstract

Self-consistency (SC) is a widely-used test-time inference technique for improving performance in chain-of-thought reasoning. It consists of generating multiple responses, or "samples," from a large language model (LLM) and selecting the most frequent answer. This procedure can naturally be viewed as a majority vote or empirical mode estimation. Despite its effectiveness, self-consistency is prohibitively expensive at scale when naively applied to datasets, and it lacks a unified theoretical treatment of sample efficiency and scaling behavior. In this paper, we provide the first comprehensive analysis of SC's scaling behavior and its variants, drawing on mode estimation and voting theory. We derive and empirically validate power law scaling for self-consistency across datasets, and analyze the sample efficiency for fixed-allocation and dynamic-allocation sampling schemes. From these insights, we introduce `Blend-ASC`, a novel variant of self-consistency that dynamically allocates samples to questions during inference, achieving state-of-the-art sample efficiency. Our approach uses $6.8\times$ fewer samples than vanilla SC on average, outperforming both fixed- and dynamic-allocation SC baselines, thereby demonstrating the superiority of our approach in terms of efficiency. In contrast to existing variants, we note that Blend-ASC is hyperparameter-free and can fit any budget of samples, ensuring it can be easily applied to any self-consistency application.

## 1 Introduction

Test-time inference has emerged as a promising direction for improving the performance of large language models (LLMs) on reasoning-intensive tasks (Wei et al., 2022; Snell et al., 2025). These techniques encourage models to "think more" by either exploring diverse reasoning paths (Yao et al., 2023) or producing longer outputs (Muennighoff et al., 2025). Among such approaches, *self-consistency* (SC) (Wang et al., 2023), also known as Vote $@\,n$, has become widely adopted due to its simplicity and efficiency. For each question, it suffices to sample $n$ chain-of-thought generations and select the most frequent answer. In other words, SC is equivalent to a plurality vote across the sampled outputs, and can be viewed as selecting the empirical mode of the LLM's answer distribution. Beyond improving the accuracy of LLMs, SC was also successfully used for preference optimization (Prasad et al., 2025) and enhancing the reliability of LLMs (Novikova et al., 2025), making it an active research topic as confirmed by numerous recent publications (Huang et al., 2024; Zhang et al., 2024; Cheng et al., 2025; Abdulaal et al., 2025).

Despite the clear ties of SC to mode estimation and voting theory, most attempts to improve or analyze it have relied on ad-hoc statistical methods or semantic approaches (Du et al., 2025; Chen et al., 2023), often overlooking insights from the rich existing literature. The absence of these fundamental approaches leaves SC and its variants without a principled analysis of their sample efficiency, as well as provable guarantees. Yet such an analysis is essential, since SC can be highly inefficient at scale. Under a fixed sampling budget, vanilla SC distributes samples uniformly across questions, regardless of their difficulty. The efficiency could be improved by allocating samples adaptively, focusing more on harder questions (Aggarwal et al., 2023; Li et al., 2024). While such adaptive variants of SC are able to dramatically enhance efficiency, they remain rather underexplored. To

address this gap, this work provides a comprehensive analysis of the sample efficiency of SC and its related variants using mode-estimation and voting theory results.

We show that SC follows power-law scaling and identify variants with accelerated and even exponential error decay. Following our analysis, we introduce `Blend-ASC`, a novel adaptive SC algorithm that achieves the best empirical sample efficiency. Motivated by existing mode estimation results, our algorithm matches the initial performance of existing SC variants in the low-sample regime and outperforms all variants at scale. To ensure ease of use for practitioners, we also adapt our method and existing methods to be hyperparameter-free. Ultimately,

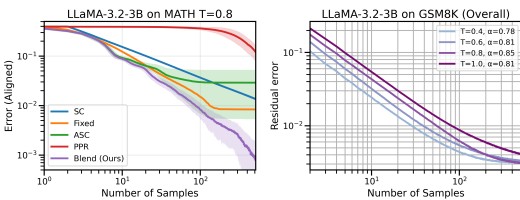

Figure 1: (Left) Scaling laws of SC established in our work and confirmed in practice. (Right) `Blend-ASC` converges to the limiting answer the fastest on aligned questions.

our algorithm leads to a significant improvement in sample efficiency, requiring $6.8\times$ fewer samples on average than vanilla SC.

**Main contributions.** Our contributions can be summarized as follows:

1. We leverage mode estimation and voting theory (Aeeneh et al., 2025; Anand Jain et al., 2022) to analyze the scaling and sample efficiency of SC, fixed-allocation SC, and dynamic-allocation SC, identifying clear power laws for (Fig. 1 (left)) SC performance and accelerated convergence for SC variants. Our theoretical results are tighter than previous work on per-question scaling (Huang et al., 2025a) and are the first to cover dynamic allocation SC.

2. We introduce `Blend-ASC`, a hyperparameter-free SC variant that achieves optimal sample-efficiency for a given budget, by combining an asymptotically-optimal algorithm with an existing adaptive variant (Fig. 1 (right)).

3. We validate our results with extensive experiments and simulations across models and benchmarks, demonstrating practical benefits.

## 2 SELF-CONSISTENCY AS MODE ESTIMATION AND MAJORITY VOTE

**Setup.** Let $q$ be an input question, fed to an LLM that outputs a chain-of-thought (CoT) yielding a final answer $r$. Let $\mu(\cdot \mid q)$ be the distribution of such answers. We say that the LLM is aligned to a question $q$ if the true response $r^*$ is the mode of the LLM distribution, i.e., $r^* = \arg\max_r \mu(r \mid q)$, and misaligned otherwise. Self-Consistency (SC) samples $x$ CoT generations $r_1, \ldots r_x \sim \mu(r \mid q)$ and the output is the most frequent answer. In other words, an empirical distribution $\hat{\mu}_x$ is generated based on $x$ sampled answers, and the output of SC is the empirical mode $r_{SC} = \arg\max_r \hat{\mu}_x(r \mid q) = \arg\max_r \sum_{i=1}^{x} \mathbb{1}[r_i = r]$. One can notice that if the model is aligned to $q$, SC converges to the correct response as $x \to \infty$ as $\hat{\mu}_x(\cdot \mid q)$ converges to $\mu(\cdot \mid q)$. Otherwise, SC converges to an incorrect response, which implies the model is inherently not capable to answer question $q$. Thus, SC sample efficiency is measured by the rate of convergence to the true mode. From a voting theory perspective, we can view the support of $\mu$ as a list of candidates, and each response $r_i$ is a vote from i.i.d voters who select candidates with probability $\mu(\cdot \mid q)$.

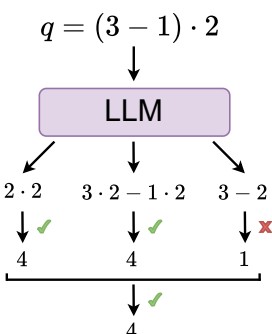

Figure 2: Example of SC.

**Per-question scaling.** To understand the non-asymptotic behavior of self-consistency, we analyze its convergence rate on a single question. Considering an aligned question $q$, we upper bound the expected error of SC defined by

$$\text{err}(x, q) = \mathbb{P}[r_{SC} \neq r^*] = \mathbb{P}\left[\arg\max_r \hat{\mu}_x(r \mid q) \neq \arg\max_r \mu(r \mid q)\right].$$

---

[1]We use $x \geq 4$ for GPQA-Diamond to have a sufficient sample size.

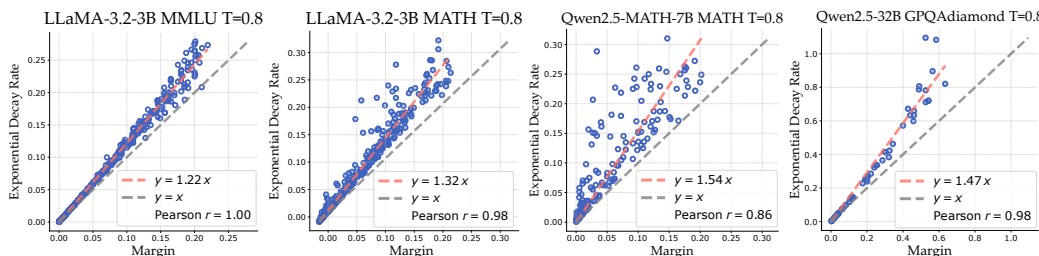

Figure 3: **Margin decay.** Margin correlates with decay rate across several model and dataset combinations, where decay is fit for $x \geq 16$ for $\epsilon$ to have negligible impact on the bound.[1]

Existing voting theory results grow unbounded as the number of candidates, or *unique* responses, increases (Aeeneh et al., 2025; Hu et al., 2024). Note that LLMs can vacuously produce infinite unique responses for a question[2]. We therefore extend the error bound in Aeeneh et al. (2025) to handle numerous unique responses. We address this by grouping the tail of low-probability responses in $\mu(\cdot \mid q)$, deriving a stronger bound by bounding $\mathbb{P}[r_{SC} \neq r^*]$ over top $k \ll K$ answers.

> **Theorem 1.** *Let $q$ be a model-aligned question. Without loss of generality, we consider the unique responses sorted according to $\mu(\cdot \mid q)$ in descending order, denoting their corresponding probabilities by $p_1 \geq p_2 \geq p_3 \geq \dots$. Let $\bar{p}_k = \sum_{i \geq k} p_i$ with $k$ that satisfies $\bar{p}_k < p_2$. Then, for the empirical distribution $\hat{\mu}_x$ from $x$ samples, the self-consistency error satisfies*
> $$\text{err}(x, q) \leq \exp(-x((\sqrt{p_1} - \sqrt{p_2})^2 + \epsilon))$$
> *with $\epsilon \to 0$ as $x \to \infty$ with rate $O\left(\frac{\log x}{x}\right)$. For misaligned $q$, the bound holds for $1 - \text{err}(x, q)$.*

The proof is deferred to Section C.1. By restricting $k$ to the smallest integer such that $\bar{p}_k < p_2$, we substantially reduce the effective number of candidates. In general, Theorem 1 demonstrates exponential error convergence with rate $m = (\sqrt{p_1} - \sqrt{p_2})^2$, which we refer to as the *margin*. The margin reflects model confidence by quantifying the gap between its most likely answer and the second most likely one. Note that Li et al. (2025) and Huang et al. (2025a) also consider the notion of margin, while in a broader range of unsupervised learning tasks, it is common to rely on empirical margin distributions, even when the estimates may be potentially noisy (Feofanov et al., 2024; Xie et al., 2024).

**Remark.** We now briefly comment on the main assumption of the theorem, which assumes that $q$ is a model-aligned question. To this end, we first note that the alignment assumption of this kind is a restriction of the self-consistency framework itself and all verifier-free parallel scaling techniques, rather than an assumption introduced artificially by our work. Indeed, without an external oracle/feedback, we cannot correct systematic errors in the underlying base model. With self-consistency, we can only amplify the existing signal, identifying the mode, and seek to do this most efficiently. Overall, we note that for well-performing models, the scaling behaviour is still dominated by model-aligned questions, and misaligned ones only represent an irreducible error.

To validate Theorem 1 empirically, we consider three recent models, LLaMA-3.2-3B, Qwen-2.5-Math, and Qwen-2.5-32B, evaluated on MMLU, MATH, and GPQA-Diamond. For each question, we estimate the margin using 100 samples. We calculate the empirical decay rate by fitting an exponential curve to the mode estimation error. Fig. 3 shows that all models' performance exhibits a clear correlation between the margin and the error decay rate with an increasing number of voting samples. This behavior is consistent across models and benchmarks as illustrated in Appendix D.

**Comparison with prior work.** Huang et al. (2025a) introduces a per-question sample-efficiency bound, stating that $x \geq 2 \log(\frac{1}{\delta})/(p_1 - p_2)^2$ many samples achieves an error of $\delta$. Our result achieves error less than or equal to $\delta$ when $x \geq \log(\frac{1}{\delta})/((\sqrt{p_1} - \sqrt{p_2})^2 + \epsilon)$. We have a tighter result when $2(\sqrt{p_1} - \sqrt{p_2})^2 + \epsilon \geq (p_1 - p_2)^2$. To show this, we note that $\epsilon \to 0$ as $x$ increases, so we consider

---

[2]For a free-response math question, the set of unique responses could be the set of integers.

the simplified bound with $\epsilon = 0$, which always holds. As $(\sqrt{p_1} - \sqrt{p_2})^2 = \frac{(p_1-p_2)^2}{(\sqrt{p_1}+\sqrt{p_2})^2}$, we have that

$$\frac{1}{(\sqrt{p_1} - \sqrt{p_2})^2} = \frac{(\sqrt{p_1} + \sqrt{p_2})^2}{(p_1 - p_2)^2} \leq \frac{2\sqrt{p_1}^2 + 2\sqrt{p_2}^2}{(p_1 - p_2)^2} \leq \frac{2}{(p_1 - p_2)^2},$$

which implies our result. This suggests that margin is a more natural measure of confidence in SC compared to the absolute difference $p_1 - p_2$. Besides the per-question bound, we provide a general analysis on the dataset setting to improve SC efficiency, which they do not explore.

## 3   SCALING LAWS ON DATASET PERFORMANCE

We broaden our analysis by extending it to the study of the sample efficiency over benchmarks and datasets, the setting in which sample efficiency can be meaningfully improved. Such a study is more informative than individual questions as it provides insights into the empirical behavior of SC when aggregated over a full dataset.

**Synthetic datasets.**    In order to study a dataset level performance theoretically, we now consider a dataset $\mathcal{D}$ of infinitely-many aligned questions, $(q_i)_{i\in\mathbb{N}}$ with margins $m_i$. Using the theoretical error model $\mathrm{err}(x, q_i) = \exp(-m_i x)$ from Theorem 1, the expected dataset error with $x$ samples for each question is

$$\mathrm{err}(x, \mathcal{D}) = \mathbb{E}_{q_i\sim\mathsf{Unif}(\mathcal{D})}\left[\mathrm{err}(x, q_i)\right] = \mathbb{E}_{q_i\sim\mathsf{Unif}(\mathcal{D})}\left[\exp(-m_i x)\right] = \int_0^1 e^{-m_i x} p_{\mathcal{D}}(m)dm$$

which is precisely the Laplace Transform $\mathcal{L}\{p_{\mathcal{D}}(m)\}$ where $p_{\mathcal{D}}(m)$ is the probability density function of margin across $\mathcal{D}$. Note that $\mathcal{L}\{p_{\mathcal{D}}(m)\}$ scales as $x^{-1/2}$ if $p(m) \propto m^{-1/2}$ and scales as a power law if $p(m)$ does as well and the exponent is greater than $-1$. We prove that we only need $p_{\mathcal{D}}(m) \propto m^{-1/2}$ near 0 to have $x^{-1/2}$ scaling (see Theorem 5 in Appendix). Then, since our error model is only dependent on margin, $m = (\sqrt{p_1} - \sqrt{p_2})^2$, we show that margin naturally leads to power law scaling and often encourages $p_{\mathcal{D}}(m) \propto m^{-1/2}$ near 0 for several constructive examples of families of datasets $\mathcal{D}$ defined as follows:

$\mathcal{D}_1$. Distribution of top two probabilities $(p_1, p_2)$ is uniform across $A = \{(x, y) \mid 0 \leq y \leq x \leq 1, x + y \leq 1\}$, i.e., $g(p_1, p_2) : A \to \mathbb{R}_{\geq 0}$, $g = \mathsf{Unif}(A)$.

$\mathcal{D}_2$. Distribution of $(p_1, p_2)$ is weighted by $(p_1 + p_2)^n$ for $n > 0$, ie, $g(p_1, p_2) \propto (p_1 + p_2)^n$. This arbitrarily downweights questions where both $p_1$ and $p_2$ are low, which are questions where the model has low confidence and considers several responses.

$\mathcal{D}_3$. Distribution $g(p_1, p_2) = (\sqrt{p_1} - \sqrt{p_2})^{2n}$ for $n > 0$. We refer to this case as adversarial, as it arbitrarily down-weights low-margin questions to encourage faster convergence. Regardless, $\mathcal{L}\{p_{\mathcal{D}}(m)\} \propto m^{-n-1/2}$ has power-law scaling.

We now state our main result for this broad family of datasets below.

> **Proposition 2.** *For a broad class of datasets $\{\mathcal{D}_1, \mathcal{D}_2\}$, the margin distribution satisfies $\lim_{m\to 0^+} p(m) \propto \frac{1}{\sqrt{m}}$, and for adverserial dataset $\mathcal{D}_3$, $\lim_{m\to 0^+} p(m) \propto m^{-n-1/2}$.*

We illustrate these results in Fig. 4 where we show the theoretical error decay for the three dataset families considered above. We observe power law scaling in both margin and error decay, consistent with what our theory predicts. As expected, the adversarial dataset $\mathcal{D}_3$ exhibits a faster convergence due to its favorable reweighting of low-confidence questions. To show that models also naturally have power-law scaling in margin, we sample up to 100 questions from each dataset and apply a KDE to "simulate" a continuous distribution. These synthetic datasets have margin distributions remarkably close to those observed for datasets $\mathcal{D}_1$ and $\mathcal{D}_2$. This suggests that our theoretical setup is realistic enough to provide insights about the performance of SC in real-world benchmarks.

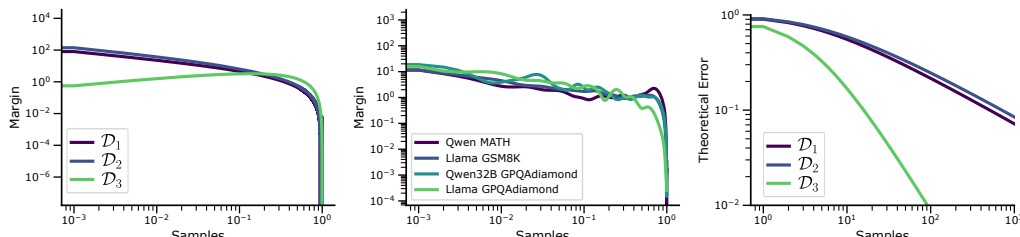

Figure 4: **Large dataset sizes induce power-law scaling.** (Left) Margin distribution for $\mathcal{D}_1 - \mathcal{D}_3$ with $n = 1$. (Middle) Margin distribution from sampling 100 points from each real-world dataset and applying KDE; (Right) Error scaling $\mathcal{D}_1 - \mathcal{D}_3$, with $\mathcal{D}_3$ having the fastest convergence.

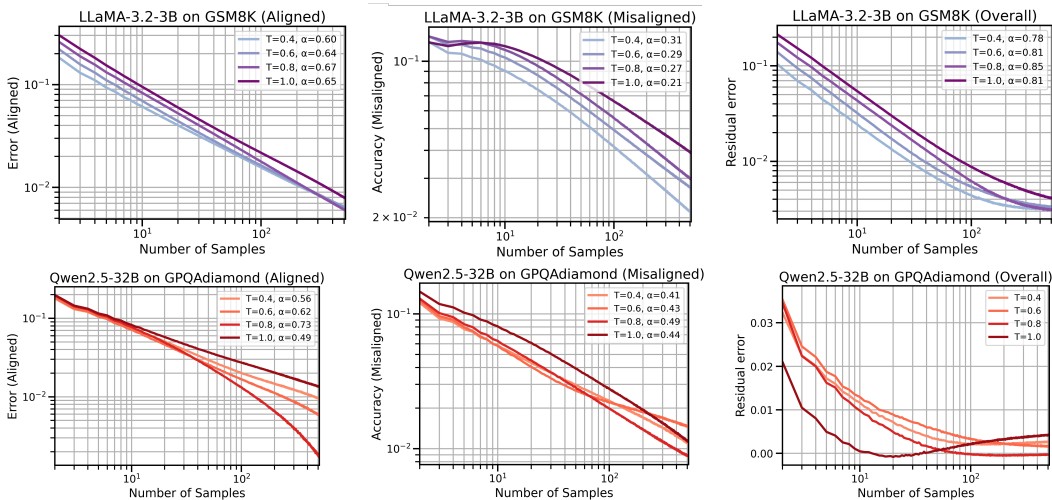

Figure 5: **Scaling behavior of Self-Consistency:** on aligned (left), misaligned (middle), and full (right) datasets for free-response (top) and multiple-choice (bottom) benchmarks.

**Empirical results.** Finally, we provide a more nuanced illustration of the observed scaling law on real-world datasets in Fig. 5. We plot the true error rate across multiple-choice and free-response benchmarks with Llama-3.2-3B on GSM8K and Qwen-32B on GPQAdiamond across temperatures ranging from 0.4 to 1. In each plot and for each temperature, we report the slope of the scaling law as $\alpha$. Our first observation is that for aligned questions (left), we have extremely strong power-law scaling, with weaker power-law scaling for misaligned questions. This can be attributed to the fact that Theorem 1 is not as strong for misaligned questions as mentioned in the proof.

For full datasets, we combine both contributions from both aligned and misaligned questions. In the case of free-response questions, we observe a consistent power-law scaling (top row). As for the multiple-choice questions, the behaviour is often non-monotic (bottow row). We hypothesize that this is because free-response questions distribute over many wrong answers, while multiple-choice questions concentrate on a few wrong answers, inflating the proportion of misaligned questions. The latter behaviour is also well-aligned with the observations made by Schaeffer et al. (2025) who showed that the test-time inference performance scaling in the case of multiple-choice tasks is difficult to predict. Chen et al. (2024) also reported that SC can even hurt performance in multiple-choice tasks as aligned questions converge to complete accuracy while misaligned questions converge to certain error or ties. Additional graphs are in Section E.

## 4 OPTIMAL ADAPTIVE SELF-CONSISTENCY

A major drawback of SC when used on datasets is sample efficiency. Some questions only need a few samples, while others require hundreds, yet we use the same number of samples per question for the whole dataset. This motivates *adaptive SC*, which allocates samples per question given some budget of total samples. We consider two settings: *fixed allocation* where samples are allocated a

priori and *dynamic allocation* where samples are allocated during inference. Fixed methods can be used when there is information about questions. Dynamic methods rely on consistency in the sample empirical distribution to inform mode stability.

## 4.1 FIXED ALLOCATION

Consider any setting where we a priori have information on questions that may inform model performance, such as question difficulty or subject. Then we may adjust the number of samples for the question based on these attributes, such as, for example, (Wang et al., 2025b) that allocates one sample for "easy" questions . To quantify the sample efficiency of fixed allocation, we consider the optimal setting where we have full information on the question, or oracle access to $\mu$.

We assume that $\text{err}(x, q_i) = \exp(-m_i x)$ and let $x_{q_i}$ be the number of samples allocated to question $q_i$ and $\bar{x}$ be the average samples per question. Then we have $\text{err}(\bar{x}, \mathcal{D}) = \mathbb{E}_{q_i \sim \text{Unif}(\mathcal{D})}[\exp(-m_i x_{q_i})]$. Since error only depends on margin, oracle access to $\mu$ means that we can access the margin $m_i$ for any $q_i$. So based on $m_i$, we should optimally choose $x_{q_i}$. We see that all questions with the same margin should have the same number of samples, so we can let $x_{q_i} = x_{m_i}$ be a function of margin. Suppose now that we have any two questions $q_i$ and $q_j$ with the same margin $m_i = m_j$. The total error is $\exp(-m_i x_{q_i}) + \exp(-m_i x_{q_j}) \geq 2\sqrt{\exp(-m_i x_{q_i})\exp(-m_i x_{q_j})} = 2\exp(-m_i(x_{q_i} + x_{q_j})/2)$. So it is optimal to distribute samples equally among questions with the same margin. Then we can define $x_m \in M \subset \{f \mid f : (0, 1] \to \mathbb{N}\}$ where $M$ is the set of functions from $(0, 1]$ to $\mathbb{N}$ such that $\bar{x} = \int_0^1 x_m p(m)dm$ for some average number of samples $\bar{x}$. We can express the error as

$$\text{err}(x_m, \mathcal{D}) = \min_{x_m \in M} \mathbb{E}_{q_i \sim \text{Unif}(\mathcal{D})}[\exp(-m_i x_{m_i})] = \min_{x_m \in M} \int_0^1 \exp(-m x_m) p(m)dm$$

which becomes a constrained convex optimization problem. For a tractable, closed-form solution, we weaken our assumption to have $x_i \geq 0$ and solve this problem by means of the following proposition.

**Proposition 3.** *Under the above assumptions and with $p(m) \propto m^{-r}$ for $r \in (0, 1)$, the optimal sample allocation is*

$$x_m = \begin{cases} m^{-1}(\log m - \log \lambda) & \text{if } m \geq \lambda \\ 0 & \text{if } m < \lambda \end{cases}$$

*where as $\bar{x} \to \infty$, $\lambda \sim \bar{x}^{-\frac{1}{r}}$. This gives us an error convergence rate of approximately $\bar{x}^{-\frac{1-r}{r}}$ which becomes $\bar{x}^{-1}$ in the special case of $r = \frac{1}{2}$*

The intuition behind this sample allocation is that for a sufficiently small margin ($m < \lambda$), it is no longer efficient to allocate any samples. This is because the marginal improvement of a single sample is less than adding a sample to a question with a higher margin. We illustrate our obtained results and the above-mentioned intuition of optimal fixed allocation in Fig. 6. We use the error model $e^{-m_i x_{m_i}}$ with margins ex-

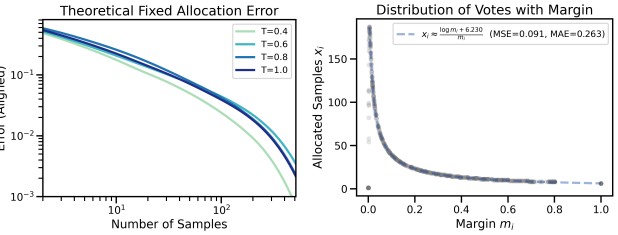

Figure 6: **Optimal allocation.** (Left) Fixed-Allocation SC scaling. (Right) The number of allocated samples closely follows the theoretical distribution depending on the margin.

tracted from running Llama-3.2-3B on MATH. The obtained scaling law across different temperatures confirms the accuracy of our theoretical result in the large sample regime. Finally, Fig. 6 (right) shows how the fixed allocation sampling works depending on the margin: low-margin questions where the LLMs is uncertain of the answer get more samples (margin is close to 0 on the x-axis), while the number of samples for high-margin (rightmost part of the x-axis) questions converges quickly to 1.

## 4.2 DYNAMIC ALLOCATION

Instead of assigning a fixed number of samples a priori for each question, dynamic allocation adaptively samples during inference until a stopping criterion $\mathcal{S}_\delta$ indicates that the self-consistency output $r_{\text{SC}}$ achieves high confidence. Following Shah et al. (2020), these criteria require at least $x(\mathcal{S}_\delta, q_i) = \Omega(\ln 1/\delta)$ samples for question $q_i$, where $\delta \in (0, 1)$ denotes the target expected error.

Existing adaptive variants such as Adaptive SC (ASC, Aggarwal et al., 2023) and Early-Stopping SC (ESC, Li et al., 2024) achieve state-of-the-art sample efficiency, substantially reducing the number of samples compared to vanilla SC. However, they are difficult to use as they require extensive parameter tuning and cannot be reliably used in settings with fixed budget constraints. Additionally, their allocation schemes are not grounded in a solid theory and rely on stopping conditions derived from heuristics. In this section, we introduce hyperparameter-free variants and provide the first theoretical analysis of the optimal dynamic allocation within the SC framework.

**Stopping criterion: PPR-1v1.** By looking at self-consistency from the mode estimation perspective, we build upon results derived by Anand Jain et al. (2022) for martingale confidence sequences. We introduce PPR-1v1 stopping criterion that we describe below. At each iteration, we allocate a sample to the question with the lowest confidence, evaluated by $(K-1)\text{Beta}(x, n_1+1, n_2+1)$ where $K$ refers to the number of possible *unique* answers, and $n_1$ and $n_2$ are counts for the two most frequent answers. For target error $\delta$, the stopping criterion is defined as $\text{Beta}(\frac{1}{2}, n_1+1, n_2+1) \leq \frac{\delta}{K-1}$.

**Asymptotically optimal allocation.** A key property of the PPR-1v1 stopping criterion is that it has a theoretically optimal *exponential* decay in error for predicting the mode since it converges to $\Theta(\ln \frac{1}{\delta})$ samples as $\delta \to 0$. In Corollary 4, we extend this result to the dataset setting.

> **Corollary 4.** *Given a dataset $\mathcal{D}$, a target error $\delta \in (0, 1)$ and stopping condition PPR-1v1 denoted as $\mathcal{S}_\delta$, we have that*
>
> $$\lim_{\delta \to 0^+} \mathbb{E}_{q_j \sim \text{Unif}(\mathcal{D})} \left[ \frac{x(\mathcal{S}_\delta^P, q_j)}{\text{LB}(\delta, q_j)} \right] = 1,$$
>
> *where $x(\mathcal{S}_\delta, q_i) \geq \sup\limits_{\rho: \arg\max(\rho) \neq \arg\max \mu(\cdot \mid q)} \frac{1}{\text{KL}(\mu(\cdot \mid q), \rho)} \ln\left(\frac{1}{2.4\delta}\right) := \text{LB}(\delta, q_i)$ and $\rho$ is a categorical distribution with the same support as $\mu(\cdot \mid q)$.*

Thus, Corollary 4 establishes the theoretically optimal decay rate for dynamic allocation and shows that the PPR-1v1 algorithm achieves this rate. The latter is also supported empirically in the large sample regime, as can be seen in Fig. 8. Thus, the introduced PPR-1v1 is an asymptotically optimal stopping condition, which makes it distinguished from the other methods used in ASC and ESC.

**Our approach: `Blend-ASC`.** Despite the theoretical guarantees for the PPR-1v1 method, we have empirically found that in the low-sample regime it tends to be pessimistic, and simpler policies like ASC are very efficient. This motivated us to introduce `Blend-ASC` that combines the best of the two worlds. For each question $Q$, we recover the confidence score from ASC and PPR-1v1 with associated rankings $\text{ASC}(Q)$ and $\text{PPR}(Q)$, respectively. Then, at step $t$ out of $T$ total samples, we generate a response for the question that minimizes the linearly-scaled ranking $(1 - \frac{t}{T})\text{ASC}(Q) + \frac{t}{T}\text{PPR}(Q)$. This is summarized in Fig. 7. We also observe that PPR-1v1 overly concentrates samples on a few questions, so we exclude questions with over 16 times the number of samples than the average question.

**Hyperparameter-free and fixed budget.** For each question, ASC and ESC sample until they reach a stopping criterion defined by input hyperparameters, and then move to the next question. This results in a variable number of samples per instance, as allocations are stochastic across questions. In contrast, we jointly allocate samples across all questions. We reformulate our stopping criterion into a parameter-free confidence score, and assign samples to questions with low confidence. See more details in Section A and Section B.

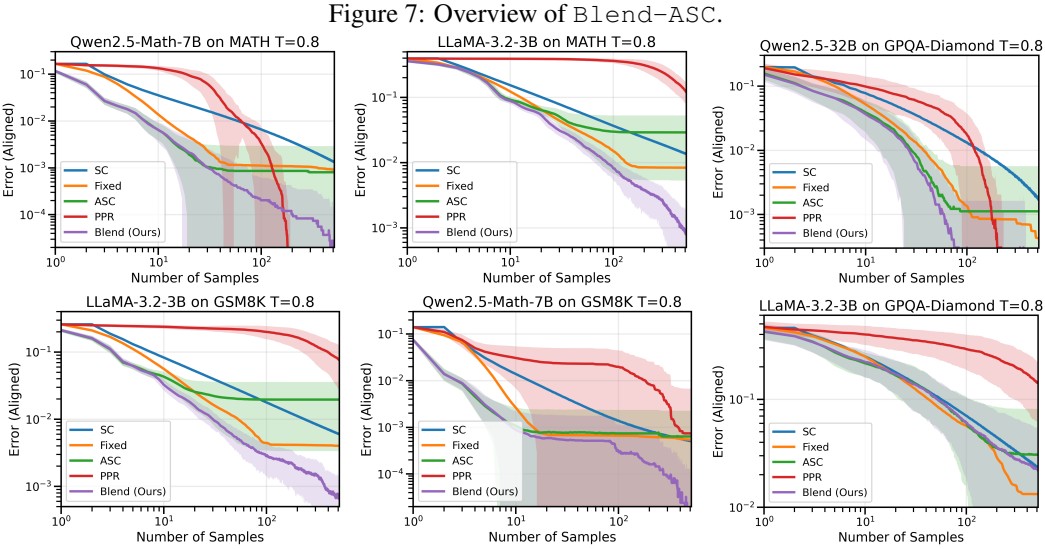

Figure 7: Overview of `Blend-ASC`.

Figure 8: **Performance comparison.** `Blend-ASC` consistently outperforms all methods in mode-estimation across models and datasets, achieving the lowest sample efficiency for target error.

Developing hyperparameter-free methods is an important improvement over existing methods. First, both ASC and ESC require tuning two hyperparameters, increasing the difficulty of implementation as users need to optimize for efficiency and cost. In contrast, our method is lightweight and can directly reach a desired budget. It iteratively increases the confidence for each question, so we can evaluate our algorithm's performance during runtime to prematurely stop it after it has converged.

**Related work.** We note that despite extensive adaptations to SC (Wang et al., 2025a; Taubenfeld et al., 2025), there are few papers analyzing SC behavior and sample efficiency both empirically and theoretically. Chen et al. (2024) showed that there is often no monotonic increase in SC performance with samples on multiple choice benchmarks. Ruan et al. (2024) uses "observational" scaling laws which fit curves across several LLMs to predict SC behavior, but observe a weak scaling trend with FLOPs. From a theoretical point of view, Hu et al. (2024) introduced a per-question bound on error, but the bound scales with the number of unique reasoning steps, which can be vacuously large. Huang et al. (2025a) provide a per-question bound similar to Theorem 1 with detailed comparisons in the dedicated section, but they are limited to the per-question setting and do not explore improving efficiency. Finally, we note that several other extensions to SC were recently proposed in the research community that use the LLM to reflect on the initially drawn responses in order to rectify them if needed. Some representative examples of these include mirror-consistency (Huang et al., 2024), self-check (Miao et al., 2023), and self-contrast (Zhang et al., 2024).

## 5    NUMERICAL EXPERIMENTS

**Benchmarks and models.**    We evaluate our findings using a variety of models, including LLaMA-3.2-3B, Qwen2.5-MATH-7B, and Qwen2.5-32B, and temperature settings from 0.4 to 1.0. We

Table 1: **Sample efficiency of adaptive methods.** We compare how many samples are required to achieve a lower accuracy on aligned questions than SC using 64 and 128 samples. ASC doesn't reach the target SC accuracy for Llama on GSM8K, so we let the samples be 128.

| SC@n | Algorithm | GSM8K | | MATH | | GPQA-Diamond | | Average Improvement |
|---|---|---|---|---|---|---|---|---|
| | | Llama-3B | Qwen-Math | Llama-3B | Qwen-Math | Llama-3B | Qwen-32B | |
| 64 | Fixed-Allocation | 15 | 9 | 18 | 10 | 27 | 15 | 4.66× |
| | Adaptive SC | 13 | 6 | 16 | 7 | 33 | 13 | 5.93× |
| | **Blend-ASC (Ours)** | 11 | 6 | 14 | 7 | 26 | 8 | **6.78×** |
| 128 | Fixed-Allocation | 34 | 14 | 45 | 19 | 77 | 37 | 4.60× |
| | Adaptive SC | 128 | 11 | 77 | 13 | 102 | 29 | 4.97× |
| | **Blend-ASC (Ours)** | 22 | 9 | 31 | 12 | 80 | 25 | **6.92×** |

(a) Sample efficiency at temperature 0.8.

| SC@n | Algorithm | GSM8K | | MATH | | GPQA-Diamond | | Average Improvement |
|---|---|---|---|---|---|---|---|---|
| | | Llama-3B | Qwen-Math | Llama-3B | Qwen-Math | Llama-3B | Qwen-32B | |
| 64 | Fixed-Allocation | 23 | 17 | 25 | 11 | 53 | 23 | 2.53× |
| | Adaptive SC | 21 | 10 | 31 | 9 | 49 | 30 | 2.56× |
| | **Blend-ASC (Ours)** | 16 | 9 | 20 | 8 | 49 | 21 | **3.12×** |
| 128 | Fixed-Allocation | 36 | 21 | 41 | 15 | 82 | 37 | 3.31× |
| | Adaptive SC | 128 | 13 | 68 | 13 | 87 | 43 | 2.18× |
| | **Blend-ASC (Ours)** | 25 | 12 | 29 | 10 | 90 | 26 | **4.00×** |

(b) Sample efficiency at temperature 1.0.

evaluate on both free response and multiple choice datasets including GSM8K, MATH, MMLU, and GPQA-Diamond. Due to the high computational cost of scaling test-time inference, directly running inference can be prohibitively expensive in time and compute. So, we sample 100 generations per question from an LLM to form the "true" LLM distribution. SC is performed by sampling from the corresponding multinomial distribution. We focus on aligned questions as done in Huang et al. (2025a). This allows a fair baseline as fixed allocation SC is not appropriate for misaligned questions as it would allocate a single sample to misaligned questions.

**Baselines.** We use hyperparameter-free ASC and PPR-1v1 as dynamic allocation baselines, each run 100 times per model and benchmark pair. Fixed allocation SC is intractable as our problem is a non-convex integer programming problem, so we use a modified method. We leave implementation details in Section B. To assess sample efficiency of SC variants, we calculate the error of SC at 64 and 128 samples, and then identify the least number of average samples required to match the error for each variant. If a method does not reach the desired error, we set the samples used to 64 or 128.

**Results.** In both Fig. 8 and Table 1, we observe that `Blend-ASC` consistently achieves optimal performance. SC has reliable performance improvements, but performs worse on the low-sample regime. Fixed-Allocation SC performs strongly, reducing samples by 4.6 times compared to SC. But it is only competitive with the unreasonable assumption of complete oracle access to $\mu$, which is never available, and underperforms ASC and `Blend-ASC`. This suggests that allocating samples before inference using prior information is never optimal. With our parameter-free and fixed budget modifications, dynamic allocation is also easier to use than allocating samples beforehand.

For dynamic allocation, PPR-1v1 is the worst method in the low-sample regime but can beat ASC in the large-sample regime, as its theoretical exponential performance suggests. ASC performs strongly in the low-sample regime but often gets stuck at large samples, highlighting its weakness in having no theoretical convergence guarantees. Finally, `Blend-ASC` matches ASC in the low-sample regime but dominates as we scale up samples, reducing the number of prompts by 6.8 times compared to SC. These performance gains highlight the usefulness of the provided theoretical analysis in practice. Finally, we note that the exponential error decay suggests that we can push the number of drawn samples even further. This is highlighted by the fact that `Blend-ASC` curves in almost all plots in Fig. 8 are not plateauing even for as many as $10^3$ samples. This can be particularly suited for public model releases where even the slightest performance gain at the expense of a drastically larger compute budget is acceptable.

Table 2: **Extended analysis.** For each pair of model and dataset from Table 1 (18 pairs), we count the number of samples that `Blend-ASC` uses to achieve the same accuracy as SC at 64 or 128 samples on aligned questions. The performance reported is the average improvement in terms of sample efficiency over the 18 pairs of model-dataset. For both extensions, we observe strong performance for each temperature $T \in \{0.4, 0.6, 0.8, 1.0\}$.

Table 3: Batched prompts.

| SC@n | Batch Size | Average Improvement | | | |
|---|---|---|---|---|---|
| | | $T = 0.4$ | $T = 0.6$ | $T = 0.8$ | $T = 1.0$ |
| 64 | 1 | 5.76 | 5.22 | 5.08 | 4.44 |
| | 2 | 4.79 | 4.32 | 4.70 | 4.30 |
| | 4 | 4.77 | 4.34 | 4.71 | 4.28 |
| | 8 | 5.65 | 5.86 | 4.54 | 4.17 |
| | 16 | 5.25 | 5.51 | 6.91 | 4.25 |
| | 32 | 4.91 | 4.65 | 5.79 | 4.17 |
| 128 | 1 | 8.40 | 7.37 | 6.93 | 6.56 |
| | 2 | 6.62 | 5.79 | 6.50 | 5.35 |
| | 4 | 6.72 | 5.95 | 6.34 | 5.43 |
| | 8 | 7.65 | 7.33 | 5.56 | 5.14 |
| | 16 | 6.81 | 6.71 | 6.23 | 5.19 |
| | 32 | 6.44 | 6.11 | 5.54 | 5.15 |

Table 4: Batched queries.

| SC@n | #Groups | Average Improvement | | | |
|---|---|---|---|---|---|
| | | $T = 0.4$ | $T = 0.6$ | $T = 0.8$ | $T = 1.0$ |
| 64 | 1 | 5.76 | 5.22 | 5.08 | 4.44 |
| | 2 | 5.70 | 5.24 | 4.87 | 4.43 |
| | 4 | 5.51 | 5.10 | 4.57 | 4.47 |
| | 8 | 5.09 | 4.85 | 4.61 | 4.31 |
| | 16 | 4.91 | 4.69 | 4.46 | 4.14 |
| 128 | 1 | 8.40 | 7.37 | 6.93 | 6.56 |
| | 2 | 7.84 | 7.16 | 6.53 | 6.24 |
| | 4 | 7.91 | 7.37 | 6.47 | 6.11 |
| | 8 | 7.03 | 6.42 | 6.27 | 5.73 |
| | 16 | 6.54 | 6.17 | 6.01 | 5.43 |

**Extended analysis.** Finally, we explore extensions of `Blend-ASC` for realistic deployment scenarios. In particular, we study batched aggregation of queries to improve latency and how to adapt to the online nature of queries in many real-world applications.

1. **Batched prompts.** Batching is critical to real-world throughput as it generates LLM responses in parallel. To support this, we propose a simple modification to our algorithm in Section 4.2. For a batch size $b$, we select the lowest $b$ questions that minimize the rank $(1 - \frac{t}{T})\text{ASC}(Q) + \frac{t}{T}\text{PPR}(Q)$, with $\text{ASC}(Q), \text{PPR}(Q)$ the ranking from ASC and PPR-1v1 respectively. We then run batch inference on these questions and increment our step as $t \leftarrow t + b$. From Table 3, we observe strong results across batch sizes and do not observe any clear downward trend with batch size. This suggests that `Blend-ASC` remains competitive even with batched generations.

2. **Batched queries.** Our theoretical results assume that we run `Blend-ASC` on a dataset in an offline manner. However, many situations involve servicing a stream of latency-sensitive queries instead of a fixed dataset. While an optimal solution would depend on the query arrival distribution, latency requirements, etc., we propose an adaptation of `Blend-ASC` to this case: we aggregate streamed queries until we reach a certain batch size, and process them as a group using `Blend-ASC`, each with the same budget. To satisfy latency constraints, this baseline must perform well on small batch sizes, which is the case as can be seen in Table 4. We progressively increase the number of groups, thereby decreasing the batch size, and observe steady performance. Improving this baseline to achieve robust performance in a low-latency setting is an interesting direction for future work.

# 6 CONCLUSION AND FUTURE WORK

In this work, we introduce a comprehensive framework based on mode estimation and voting theory, leading to theoretical convergence guarantees and scaling laws. We first analyze SC performance on individual questions and identify power-law scaling behavior across full datasets. With this foundation, we derived improved theoretical sample efficiency results for variants on synthetic datasets, which are validated by empirical results. Finally, we introduced `Blend-ASC`, a novel parameter-free SC variant that combines an asymptotically optimally SC variant with Adaptive SC. Experiments show that `Blend-ASC` consistently outperforms all previous methods. We believe that mode estimation and voting theory results on weighted voting can inspire similar analysis for other test-time inference methods that use a verifier or LLM to generate scores and then perform majority vote, such as Best-of-N-Weighted or Self-Calibration (Snell et al., 2025; Huang et al., 2025b). We can also leverage the mean and median estimation literature to analyze SC variants that predict continuous values, such as with time-series prediction (Liu et al., 2025). Finally, self-consistency can provide a useful signal for preference optimization as shown in (Prasad et al., 2025). Our work provides a solid theoretical foundation for making this application of SC as efficient as possible.

ETHICS STATEMENT

This paper presents work whose goal is to advance the field of Machine Learning. There are many potential societal consequences of our work, none of which we feel must be specifically highlighted here.

REPRODUCIBILITY STATEMENT

Details of our theoretical and experimental results are provided in to reproduce our work. The implementation details for the numerical experiments are given in the main paper and the appendix while the proofs are detailed in the appendix. Our code will be open-sourced upon publication.

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

# Appendix

## TABLE OF CONTENTS

## A    ADAPTIVE SC AND EARLY-STOPPING SC

ASC specifically looks at the counts of the two most frequent classes, $n_1$ and $n_2$, sampled with probabilities $p_1$ and $p_2$. It considers a Beta prior on the distribution of $\frac{p_1}{p_1+p_2}$, and retrieves its posterior distribution given observed counts $n_1$ and $n_2$. Then the stopping condition activates when the probability of $p_1 < p_2$ is lower than some fixed threshold $\tau$:

$$\mathbb{P}\left[p_1 < p_2\right] = \int_0^{1/2} \text{Beta}\left(x, n_1 + 1, n_2 + 1\right) dx < \tau$$

or we reach some maximum number of samples. We then output $n_1$.

ESC repeatedly samples windows of size $w$ and until we reach a maximum number of samples or a window where all samples have the same answer, in which ESC returns that answer. From a mode-estimation perspective, ESC is not theoretically optimal, as it should strictly return the empirical mode regardless of the last window's unique answer (though they often coincide).

Finally, they do not admit a straightforward extension to a setting with a fixed budget of samples, making them unreliable and difficult to use. The average number of samples is determined by hyperparameters, and even with the same set of parameters, the number of samples on each instance is stochastic.

## B    IMPLEMENTATION DETAILS

**Fixed allocation.**    In practice, the optimal fixed allocation for a true dataset is infeasible as the average error for a question $q$ given $x$ samples is often non-convex in $x$. Since $x$ must be integers, we are thus solving a non-convex integer programming over potentially thousands of variables (sample allocation for each question). However, we observe in Section 2 that the average error for a question highly correlates with a convex and strictly decreasing exponential upper bound.

We can closely approximate Fixed Allocation SC as a convex integer programming problem by allocating samples according to a convex and monotonic approximation, where we apply local

smoothing approaches and then fit an exponential approximation for high sample questions. Under these assumptions, for any question $q$, each additional sample yields strictly positive but diminishing improvements in error. Thus, we can greedily allocate samples at each iteration to the question with the highest marginal improvement.

**Dynamic allocation.** To develop hyperparameter-free methods, we create an array for confidences, which is measured for each question based on the consistency of the current samples for that question. Each confidence is initialized to $-\infty$ (or a small initial value). We have another array that stores the sampled responses for each question. Then, at each iteration, we choose the question with the lowest confidence to sample via a heap. After updating our counts array, we update our confidence as follows: for ASC our confidence is $\int_0^{1/2} \text{Beta}(x, n_1 + 1, n_2 + 1)dx$ and for PPR-1v1 our confidence is $(K - 1)\text{Beta}(x, n_1 + 1, n_2 + 1)$.[3] For PPR-1v1 specifically, as we don't have access to $K$ a priori, we modify $K$ to be $\min(2, \hat{K})$ where $\hat{K}$ is the number of unique answers seen thus far while sampling.

---

[3]We modify the confidence in PPR-1v1 for when $K = 1$ and when we only have $n_1 + n_2 = 1$ to avoid degenerate confidences.

## C  PROOFS

### C.1  PROOF OF THEOREM 1

We assume the distribution of answers to the question $q$, $\mu(\cdot \mid q)$, has a finite support of $n$ items. We also denote $r_{\max} = \operatorname{argmax}_r \mu(\cdot \mid q)$ (assuming that argmax outputs a single value). Then, following Theorem 3 of Aeeneh et al. (2025), for any $q$, we have

$$\mathbb{P}\left[r_{\text{SC}} \neq r_{\max}\right] \leq \exp\{-x((\sqrt{p_1} - \sqrt{p_2})^2 + \epsilon))\} = (K-1) \cdot \exp\{-x((\sqrt{p_1} - \sqrt{p_2})^2 + \hat{\epsilon}))\}$$

where $\hat{\epsilon}$ has no dependence on $K$ and ties are broken randomly (as assumed in SC). The number of unique answers $K$ can be arbitrarily large (such as the space of integers), which weakens the inequality. Nevertheless, in practice, nearly all probability mass concentrates on a few answers, so we prove that we can bound $\mathbb{P}\left[r_{\text{SC}} \neq r_{\max}\right]$ by truncating to only the top $k \ll K$ answers.

Without loss of generality, let the support of $\mu(\cdot \mid q)$ be $A = \{a_1, \ldots, a_K\}$ with $p_i = \mu(a_i \mid q)$ such that $p_1 \geq \cdots \geq p_K$ and $a_1 = r_{\max}$. Suppose $x$ answers are sampled $r_1, r_2, \ldots, r_x \sim \mu(\cdot \mid q)$ and let the count vector be $(n_1, \ldots, n_K)$ where $n_i = \sum_{j=1}^{x} \mathbb{1}[r_j = a_i]$. Define the tail bucket distribution $\tilde{\mu}$ of the original $\mu$ by aggregating all answers $a_i$ for $i \geq k$ into $\tilde{a}_k$:

$$\begin{cases} \tilde{\mu}(a_i \mid q) = \mu(a_i \mid q) & \text{if } i < k, \\ \tilde{\mu}(a_i \mid q) = \sum_{j=k}^{K} \mu(a_j \mid q) & \text{if } i = k. \end{cases}$$

Finally, by $\tilde{r}_{\text{SC}}$ we denote the self-consistency answer sampled from $\tilde{\mu}(\cdot \mid q)$.

We prove that $\mathbb{P}\left[r_{\text{SC}} \neq r_{\max}\right] \leq \mathbb{P}\left[\tilde{r}_{\text{SC}} \neq r_{\max}\right]$ by considering all possible count vectors. We can create an injection from the set of count vectors for $\mu$ into the set of count vectors for $\tilde{\mu}$ defined as $(n_1, \ldots, n_K) \to (n_1, \ldots, n_{k-1}, \sum_{j=k}^{K} n_j)$. Consider all count vectors where we could have $r_{\text{SC}} \neq r_{\max}$ (under tiebreaks). Then there exists some $i \in \{2, \ldots, K\}$ such that $n_i = \max_j n_j \geq n_1$. If $i < k$, then in the bijected vector for $\tilde{\mu}$, $n_i \geq n_1$. But if $i \geq k$, then for $\tilde{\mu}$, we have the final element is $\sum_{j=k}^{K} n_j \geq n_i \geq n_1$. If we have a strict inequality, then $n_i > n_1$ and both $r_{\text{SC}} \neq r_{\max}$ and $\tilde{r}_{\text{SC}} \neq r_{\max}$.

Consider when we have $n_i = n_1$. Now, as we break ties randomly, every count vector has some probability of failure equal to $\frac{|\arg\max_j n_j| - 1}{|\arg\max_j n_j|}$. We show that in each case, the probability of failure is the same or increases when we biject from $(n_1, \ldots, n_K) \to (n_1, \ldots n_{k-1}, \sum_{j=k}^{K} n_j)$. If $i < k$, we have three cases: either $\sum_{j=k}^{K} n_j < n_i$, $\sum_{j=k}^{K} n_j = n_i$, $\sum_{j=k}^{K} n_j > n_i$. In the first case, $|\arg\max_j n_j|$ doesn't change as for all $j \in \{k, \ldots, K\}$ we have $n_j \neq \max_j n_j$ for $\mu$ and $\sum_{j=k}^{K} n_j \neq \max_j n_j$ for $\tilde{\mu}$, so the probability of failure stays the same as $|\arg\max_j n_j|$ doesn't change. In the second case, at most one $j \in \{k, \ldots, K\}$ satisfies $n_j = \arg\max_j n_j$, but we are guaranteed $\sum_{j=k}^{K} n_j = \arg\max_j n_j$, so $|\arg\max_j n_j|$ either doesn't change or strictly increases (which increases our probability of failure). In the final case, we have complete failure for $\tilde{\mu}$. In all cases, the probability of failure stays the same or increases as desired. If $i \geq k$, either $\sum_{j=k}^{K} n_j = n_i$ or $\sum_{j=k}^{K} n_j > n_i$. In the first case, $|\arg\max_j n_j|$ stays the same, and in the latter case, we have complete failure for $\tilde{\mu}$. So we see that for all count vectors where we could have $r_{\text{SC}} \neq r_{\max}$, the probability of failure stays the same or increases for $\tilde{\mu}$, so $\mathbb{P}\left[r_{\text{SC}} \neq r_{\max}\right] \leq \mathbb{P}\left[\tilde{r}_{\text{SC}} \neq r_{\max}\right]$.

Then we can bound $\mathbb{P}\left[\tilde{r}_{\text{SC}} \neq r_{\max}\right]$ using the bound from Aeeneh et al. (2025), proving our result.

When our distribution is aligned, $r_{\max}$ is the correct answer, so $\text{err}(x, q) = \mathbb{P}\left[r_{\text{SC}} \neq r_{\max}\right]$ and we upper bound this error. When our distribution is misaligned, then to bound $1 - \text{err}(x, q)$, we notice that whenever SC predicts correctly, $r_{\text{SC}} \neq r_{\max}$ as $r_{\max}$ is an incorrect answer. Then $1 - \text{err}(x, q) \leq \mathbb{P}\left[r_{\text{SC}} \neq r_{\max}\right]$.

### C.2  PROOF OF THEOREM 2

We demonstrate that many synthetic datasets exhibit $x^{-1/2}$ scaling under self-consistency by first demonstrating that we only need $p_{\mathcal{D}}(m) \propto m^{-1/2}$ near 0 to have $x^{-1/2}$ scaling. Then, since our

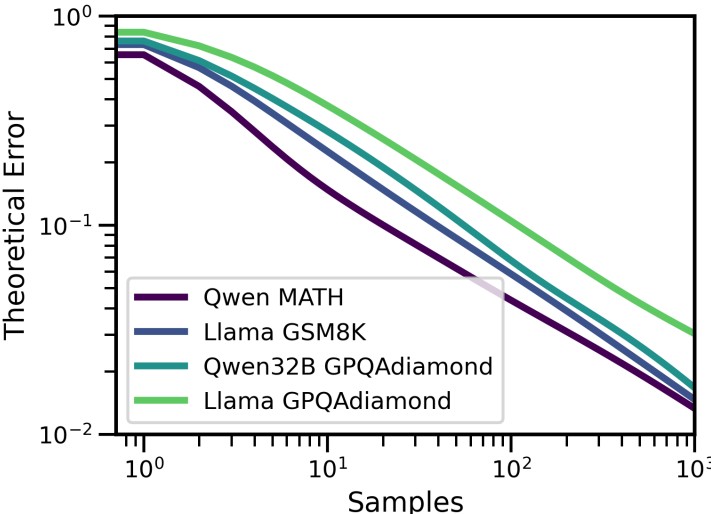

Figure 9: From Fig. 4, we observe that synthetic dataset $\mathcal{D}_1, \mathcal{D}_2, \mathcal{D}_3$ have power-law scaling under our theoretical error model. Here, we show that our kernel-smoothed distribution have approximately $x^{-1/2}$ error scaling.

error model is only dependent on margin, $m = (\sqrt{p_1} - \sqrt{p_2})^2$, we show that margin naturally encourages $p_{\mathcal{D}}(m) \propto m^{-1/2}$ near 0 by considering various synthetic datasets where the proportion of questions with top two probabilities $(p_1, p_2)$ is uniform across $A = \{(x, y) \mid 0 \leq y \leq x \leq 1, x + y \leq 1\}$ or weighted by $(p_1 + p_2)^n$. We demonstrate that even adversarially designed datasets that minimize margin also exhibit power law scaling.

The first lemma demonstrates that we only require $p(m)$ to have $m^{-1/2}$ behavior when $m \in (0, b]$ for small $b$ to get a power law lower bound on $\mathrm{err}(x, \mathcal{D})$.

**Lemma 5.** *Consider the function class*

$$F = \left\{ f(x) \in \mathcal{P}((0, 1]) \mid f(x) = \begin{cases} ax^{-1/2} & \text{if } x \leq b \\ h(x) & \text{otherwise} \end{cases}, a > 0, \forall x \ h(x) \geq 0 \right\}$$

*For any $f \in F$, $\mathcal{L}\{f\} = \frac{a}{\sqrt{x}} \gamma(\frac{1}{2}, bx) + O(e^{-bx})$ where $\gamma(\frac{1}{2}, bx) = \sqrt{\pi}(2\Phi(\sqrt{2bx}) - 1)$ is the lower incomplete gamma function and $\Phi$ is the Gaussian cumulative density function and $\Phi(x)$ converges rapidly to 1 with rate $e^{-x^2/2}$.*

*Proof.* The Laplace Transform is

$$\mathcal{L}\{f\} = \int_0^b e^{-tx} \cdot at^{-1/2}dt + \int_b^1 e^{-tx}h(t)dt$$

$\int_b^1 e^{-tx}h(t)dt$ is $O(e^{-bx})$ as the integral of $h$ must be bounded for $f$ to be a distribution. Then we have

$$\mathcal{L}\{f\} = a \int_0^b e^{-tx}t^{-1/2}dt + O(e^{-bx})$$

$$= a \int_0^{bx} e^{-u} \left(\frac{u}{x}\right)^{-1/2} \frac{du}{x} + O(e^{-bx})$$

$$= \frac{a}{\sqrt{x}} \int_0^{bx} e^{-u}u^{-1/2}du + O(e^{-bx}) = \frac{a}{\sqrt{x}} \gamma(\frac{1}{2}, bx) + O(e^{-bx})$$

$\square$

Next, we demonstrate that several synthetic datasets $\mathcal{D}$ have $p(m) \in f$.

> **Lemma 6.** *Let the distribution of $(p_1, p_2)$ across $\mathcal{D}$ be $g(p_1, p_2) : A \to \mathbb{R}_{\geq 0}$ where $A = \{(x, y) \mid 0 \leq y \leq x \leq 1, x + y \leq 1\}$. If $g = \mathsf{Unif}(A)$, then $p(m) = \frac{1}{3\sqrt{m}}\sqrt{2-m}^3 - \sqrt{m}\sqrt{2-m} + \frac{2m}{3}$, which implies that $\lim_{m \to 0^+} p(m) \propto \frac{1}{\sqrt{m}}$.*

*Proof.* We consider the cumulative distribution function of $m$, $F(m)$. Let $A_m = \{(x, y) \in A \mid (\sqrt{x} - \sqrt{y})^2 \leq m\}$.

$$F(m) = \iint_{(x,y) \in A_m} g(x, y) dx dy = 4 \iint_{(x,y) \in A_m} dx dy$$

We now find the area of $A_m$. Notice that $\sqrt{x} - \sqrt{y} \leq \sqrt{m}$ implies $x \leq (\sqrt{m} + \sqrt{y})^2$ and we have $x - y = m + 2\sqrt{my}$. Let $\hat{y} = x + y$ and $\hat{x} = x - y$. Then we have $y = \frac{1}{2}(\hat{y} - \hat{x})$ which gives us $\hat{x} - m \leq \sqrt{2m(\hat{y} - \hat{x})}$. So either $\hat{x} \leq m$ or $(\hat{x} - m)^2 \leq 2m(\hat{y} - \hat{x})$. For the latter case, we have $\hat{y} \geq \frac{1}{2m}\hat{x}^2 + \frac{1}{2}m$. We can express the constraints of $A$ as $\hat{y} \leq 1$ and $0 \leq \hat{y} - \hat{x} \leq \hat{y} + \hat{x} \leq 2$, the latter can be decomposed as $\hat{y} \geq \hat{x} \geq 0$. When $\hat{x} \leq m$, all points $(\hat{x}, \hat{y})$ where $0 \leq \hat{x} \leq \hat{y} \leq 1$ are sufficient, and for $\hat{x} \geq m$, we see that $\frac{1}{2m}\hat{x}^2 + \frac{1}{2}m \geq \hat{x}$ so all points $\frac{1}{2m}\hat{x}^2 + \frac{1}{2}m \leq \hat{y} \leq 1$ are sufficient.

Expressing this as an integral, we have

$$\frac{1}{2}F(m) = \iint_{(\hat{x},\hat{y}) \in \Phi(A_m)} d\hat{x} d\hat{y} = \int_0^m (1 - \hat{x}) d\hat{x} + \int_m^1 \max\left(0, 1 - \frac{1}{2m}\hat{x}^2 - \frac{1}{2}m\right) d\hat{x}$$

$$= m\left(1 - \frac{m}{2}\right) + \int_m^{\sqrt{2m-m^2}} \left(1 - \frac{1}{2m}\hat{x}^2 - \frac{1}{2}m\right) d\hat{x}$$

$$= \sqrt{2m - m^2}\left(1 - \frac{m}{2}\right) - \frac{1}{2m}\int_m^{\sqrt{2m-m^2}} \hat{x}^2 d\hat{x}$$

$$= \frac{1}{2m}\sqrt{2m - m^2}^3 - \frac{1}{6m}\left(\sqrt{2m - m^2}^3 - m^3\right)$$

$$= \frac{1}{3m}\sqrt{2m - m^2}^3 + \frac{m^2}{6} = \frac{1}{3}\sqrt{m}\sqrt{2 - m}^3 + \frac{m^2}{6}$$

where $\Phi$ is the transformation $(x, y) \to (\frac{1}{2}(x - y), \frac{1}{2}(x + y))$ and we remove a factor of 2 from the Jacobian in our transformation. The second equality uses $1 - \frac{1}{2m}\hat{x}^2 - \frac{1}{2}m \leq 0$ when $\hat{x} \geq \sqrt{2m - m^2}$. Taking the derivative, we have

$$p(m) = \frac{d}{dm}F(m) = \frac{1}{3\sqrt{m}}\sqrt{2 - m}^3 + 2\sqrt{m}\sqrt{2 - m}^2 \cdot \left(-\frac{1}{2\sqrt{2 - m}}\right) + \frac{2m}{3}$$

$$= \frac{1}{3\sqrt{m}}\sqrt{2 - m}^3 - \sqrt{m}\sqrt{2 - m} + \frac{2m}{3}$$

$\square$

The uniform distribution assumption is clearly not realistic, but we claim that power scaling is natural. Consider another class of datasets with distribution of $(p_1, p_2)$ weighted by $(p_1 + p_2)^n$ for $n > 0$. This arbitrarily downweights questions where both $p_1$ and $p_2$ are low, which are questions where the model has low confidence and considers several responses. We again observe that $p(m) \propto m^{-1/2}$ when $m \to 0$.

**Lemma 7.** *Let $g(p_1, p_2) \propto (p_1 + p_2)^n$ for $n > 0$, then*

$$-m^{n+1} + (n+1)\frac{1-m}{\sqrt{2-m}\sqrt{m}} + 2^{-(n+1)}\sum_{i=0}^{n+1}\binom{n+1}{i}\frac{n+2}{2i+1}m^{n+1}$$

$$-2^{-(n+1)}\sum_{i=0}^{n+1}\binom{n+1}{i}\frac{1}{2i+1}\frac{d}{dm}\sqrt{2-m}^{2i+1}m^{n-i+3/2}$$

*which implies that $\lim_{m\to 0+} p(m) \propto \frac{1}{\sqrt{m}}$.*

*Proof.* Suppose we have $(x+y)^n$

$$F(m) = \iint_{(x,y)\in A_m} g(x,y)dxdy = \iint_{(x,y)\in A_m}\frac{1}{Z}(x+y)^n dxdy$$

for normalizing constant $Z$, and from Theorem 6, we have

$$F(m) \propto \int_0^m \int_m^1 \hat{y}^n d\hat{y}d\hat{x} + \int_m^{\sqrt{2m-m^2}}\int_{\frac{1}{2m}\hat{x}^2+\frac{1}{2}m}^1 \hat{y}^n d\hat{y}d\hat{x}$$

Equivalently, we have

$$F(m) \propto \int_m^{\sqrt{2m-m^2}}\left(1 - \left(\frac{1}{2m}\hat{x}^2 + \frac{1}{2}m\right)^{n+1}\right)d\hat{x} + \int_0^m 1 - m^{n+1}d\hat{x}$$

$$= \sqrt{2m-m^2} - (2m)^{-(n+1)}\sum_{i=0}^{n+1}\binom{n+1}{i}m^{2(n+1-i)}\int_m^{\sqrt{2m-m^2}}\hat{x}^{2i}d\hat{x} - m^{n+2}$$

$$= \sqrt{2m-m^2} - 2^{-(n+1)}m^{-(n+1)}\sum_{i=0}^{n+1}\binom{n+1}{i}\frac{m^{2(n+1-i)}}{2i+1}\left(\sqrt{2m-m^2}^{2i+1} - m^{2i+1}\right) - m^{n+2}$$

$$= \sqrt{2m-m^2} - 2^{-(n+1)}\sum_{i=0}^{n+1}\binom{n+1}{i}\frac{1}{2i+1}\left(\sqrt{2-m}^{2i+1}m^{n-i+3/2} - m^{n+2}\right) - m^{n+2}$$

Taking the derivative, we have

$$p(m) = \frac{d}{dm}F(m) \propto -m^{n+1} + (n+1)\frac{1-m}{\sqrt{2-m}\sqrt{m}} + 2^{-(n+1)}\sum_{i=0}^{n+1}\binom{n+1}{i}\frac{n+2}{2i+1}m^{n+1}$$

$$-2^{-(n+1)}\sum_{i=0}^{n+1}\binom{n+1}{i}\frac{1}{2i+1}\frac{d}{dm}\sqrt{2-m}^{2i+1}m^{n-i+3/2}$$

ro We have $\frac{1}{\sqrt{m}}$ scaling (up to constant) when $m \to 0$. The first and third term decay to 0 as $m \to 0$ from the $m^{n+1}$ term. For the last term

$$\frac{d}{dm}\sqrt{2-m}^{2i+1}m^{n-i+3/2} = (n+i-\frac{3}{2})\sqrt{2-m}^{2i+1}m^{n-i+1/2} - \frac{2i+1}{2\sqrt{2-m}}\sqrt{2-m}^{2i}m^{n-i+3/2}$$

As $2 - m$ approaches 2 as $m \to 0$ the and $i$ is at most $n + 1$, the first term is at best on the order of $m^{-1/2}$ (again giving us $\frac{1}{\sqrt{m}}$ scaling), while the second term is at best on the order of $m^{1/2}$ which decays to 0. $\square$

Suppose we could observe an even faster convergence. Since questions with low margin converge the slowest, such a dataset should heavily downweight low-margin questions, so we consider $g(p_1, p_2) = (\sqrt{p_1} - \sqrt{p_2})^{2n}$ for $n > 0$. We can attempt to arbitrarily down-weight $p_1 - p_2$ by increasing $n$. However, we still have power law scaling, giving us the desired predictable gains.

**Lemma 8.** *Let $g(p_1, p_2) \propto (\sqrt{p_1} - \sqrt{p_2})^{2n}$ for $n > 0$, then*

$$F(m) \propto -m^{n+1} + (n+1)\frac{1-m}{\sqrt{2-m}\sqrt{m}} + 2^{-(n+1)} \sum_{i=0}^{n+1} \binom{n+1}{i} \frac{n+2}{2i+1} m^{n+1}$$

$$-2^{-(n+1)} \sum_{i=0}^{n+1} \binom{n+1}{i} \frac{1}{2i+1} \frac{d}{dm} \sqrt{2-m}^{2i+1} m^{n-i+3/2}$$

*which implies that $\lim_{m \to 0^+} p(m) \propto \frac{1}{\sqrt{m}}$.*

*Proof.* Suppose we have $g(x, y) = (\sqrt{x} - \sqrt{y})^{2n}$

$$F(m) = \iint_{(x,y) \in A_m} g(x, y) dx dy = \iint_{(x,y) \in A_m} \frac{1}{Z}(x + y - 2\sqrt{xy})^n dx dy$$

for some normalizing factor $Z$ and from Theorem 6, we have

$$F(m) \propto \underbrace{\int_0^m \int_m^1 \left(\hat{y} - \sqrt{\hat{y}^2 - \hat{x}^2}\right)^n d\hat{y} d\hat{x}}_{I_1(m)} + \underbrace{\int_m^{\sqrt{2m-m^2}} \int_{\frac{1}{2m}\hat{x}^2 + \frac{1}{2}m}^1 \left(\hat{y} - \sqrt{\hat{y}^2 - \hat{x}^2}\right)^n d\hat{y} d\hat{x}}_{I_2(m)}$$

We have $p(m) = \frac{d}{dm} F(m) \propto \frac{d}{dm} I_1(m) + \frac{d}{dm} I_2(m)$. Using the Leibniz Integral rule, we have

$$\frac{d}{dm} I_1(m) = \int_m^1 \left(\hat{y} - \sqrt{\hat{y}^2 - m^2}\right)^n d\hat{y} + \int_0^m \frac{\partial}{\partial m} \int_m^1 \left(\hat{y} - \sqrt{\hat{y}^2 - x^2}\right)^n d\hat{y} d\hat{x}$$

$$= \int_m^1 \left(\hat{y} - \sqrt{\hat{y}^2 - m^2}\right)^n d\hat{y} - \int_0^m \left(m - \sqrt{m^2 - x^2}\right)^n d\hat{x}$$

and

$$\frac{d}{dm} I_2(m) = \frac{1-m}{\sqrt{2m-m^2}} \int_1^1 \left(\hat{y} - \sqrt{\hat{y}^2 - 2m + m^2}\right)^n d\hat{y} d\hat{x} - \int_m^1 \left(\hat{y} - \sqrt{\hat{y}^2 - m^2}\right)^n d\hat{y}$$

$$+ \int_m^{\sqrt{2m-m^2}} \frac{\partial}{\partial m} \int_{\frac{1}{2m}\hat{x}^2 + \frac{1}{2}m}^1 \left(\hat{y} - \sqrt{\hat{y}^2 - \hat{x}^2}\right)^n d\hat{y} d\hat{x}$$

$$= -\int_m^1 \left(\hat{y} - \sqrt{\hat{y}^2 - m^2}\right)^n d\hat{y} + \int_m^{\sqrt{2m-m^2}} \frac{1}{2}\left(\frac{\hat{x}^2}{m^2} - 1\right)\left(\frac{\hat{x}^2}{2m} + \frac{m}{2} - \sqrt{(\frac{\hat{x}^2}{2m} + \frac{m}{2})^2 - \hat{x}^2}\right)^n d\hat{x}$$

$$= -\int_m^1 \left(\hat{y} - \sqrt{\hat{y}^2 - m^2}\right)^n d\hat{y} + \frac{m^n}{2} \int_m^{\sqrt{2m-m^2}} \left(\frac{\hat{x}^2}{m^2} - 1\right) d\hat{x}$$

We then have

$$p(m) \propto -\int_0^m \left(m - \sqrt{m^2 - x^2}\right)^n d\hat{x} + \frac{m^n}{2} \int_m^{\sqrt{2m-m^2}} \left(\frac{\hat{x}^2}{m^2} - 1\right) d\hat{x}$$

$$= -m \int_0^{\pi/2} \left(m - \sqrt{m^2 - m^2 \sin^2 \theta}\right)^n \cos\theta d\theta + \frac{m^n}{2} \left(\frac{2}{3}m - \sqrt{2m - m^2} + \frac{\sqrt{2m - m^2}^3}{3m^2}\right)$$

$$= -m^{n+1} \int_0^{\pi/2} (1 + \cos\theta)^n \cos\theta d\theta + \frac{m^{n-1}}{3}\left(m^2 + (1 - 2m)\sqrt{2m - m^2}\right)$$

For the first term, we define $\theta$ via $\hat{x} = m\sin\theta$ As $m \to 0$, we see that the very last term dominates and scales as $m^{n-1/2}$. Functions proportional to $m^{n-1/2}$ have Laplace Transforms that are also power laws with rate $m^{-n-1/2}$.

$\square$

## C.3 Proof of Theorem 3

The fixed allocation objective is

$$\min \int_0^1 \exp(-m x_m) p(m) dm$$

under the constraint that $\int_0^1 x_m dm = \bar{x}$ and $x_m \in \mathbb{N}$. We make the simplifying assumption that $x(m) \in [0, \infty)$ to avoid integer programming. As our error models, $\exp(-m x_m)$, are smooth and monotone, this continuous relaxation only changes results by a negligible rounding error.

Under Lagrangian optimality, the solution is of the following form:

$$x_m = \begin{cases} m^{-1}(\log m - \log \lambda) & \text{if } m \geq \lambda \\ 0 & \text{if } m < \lambda \end{cases}$$

for some $\lambda > 0$. The budget constraint then becomes

$$\bar{x} = \int_\lambda^1 (\log m - \log \lambda) \frac{p(m)}{m} dm$$

and the error becomes

$$\text{err}(\bar{x}, \mathcal{D}) = \int_0^\lambda p(m) dm + \lambda \int_\lambda^1 \frac{p(m)}{m} dm$$

Suppose the distribution of margin is of the form $p(m) = (1 - \alpha) \cdot m^{-\alpha}$ for some $\alpha \in (0, 1)$. Then the budget constraint is

$$\bar{x} = (1 - \alpha) \int_\lambda^1 m^{-1-\alpha} \log m \, dm - (1 - \alpha) \log \lambda \int_\lambda^1 m^{-1-\alpha} dm$$

$$= \frac{1-\alpha}{\alpha} \left[ m^{-\alpha} \log m \right]_1^\lambda + \frac{1-\alpha}{\alpha} \int_\lambda^1 m^{-1-\alpha} dm - (1-\alpha) \log \lambda \int_\lambda^1 m^{-1-\alpha} dm$$

$$= \frac{1-\alpha}{\alpha^2} \lambda^{-\alpha} - \frac{1-\alpha}{\alpha^2} + \frac{1-\alpha}{\alpha} \log \lambda$$

As $\bar{x} \to \infty$, we see that $\lambda$ scales proportional to $\bar{x}^{-\frac{1}{\alpha}}$. We can also express the error for

$$\text{err}(\bar{x}, \mathcal{D}) = (1 - \alpha) \int_0^\lambda m^{-\alpha} dm + (1 - \alpha)\lambda \int_\lambda^1 m^{-1-\alpha} dm$$

$$= \frac{1}{\alpha} \lambda^{1-\alpha} - \frac{1-\alpha}{\alpha} \lambda$$

Again, as $\bar{x} \to \infty$, we have that our error scales as $\lambda^{1-\alpha}$ and therefore $\bar{x}^{-\frac{1-\alpha}{\alpha}}$. In the special case where $\alpha = \frac{1}{2}$, we have that error scales as $\bar{x}^{-1}$ as $\bar{x} \to 0$.

From Proposition 3, questions with very small margins, less than $\lambda$, are ignored since allocating to them is inefficient. Fig. 10 confirms the predicted allocation and convergence across synthetic datasets, and we observe strong accelerated power-law scaling behavior on benchmarks.

## C.4 Proof of Theorem 4

Anand Jain et al. (2022) showed that PPR-1v1 is asymptotically optimal per question. We briefly show that this extends to the dataset setting. First, from Shah et al. (2020), we have

**Theorem 9.** *For $\delta \in (0, 1)$ and stopping condition $\mathcal{S}_\delta$, the expected number of samples is at least*

$$x(\mathcal{S}_\delta, q_i) \geq \sup_{\rho: \arg\max(\rho) \neq \arg\max \mu(\cdot|q)} \frac{1}{\text{KL}(\mu(\cdot \mid q), \rho)} \ln\left( \frac{1}{2.4\delta} \right) := \text{LB}(\delta, q_i)$$

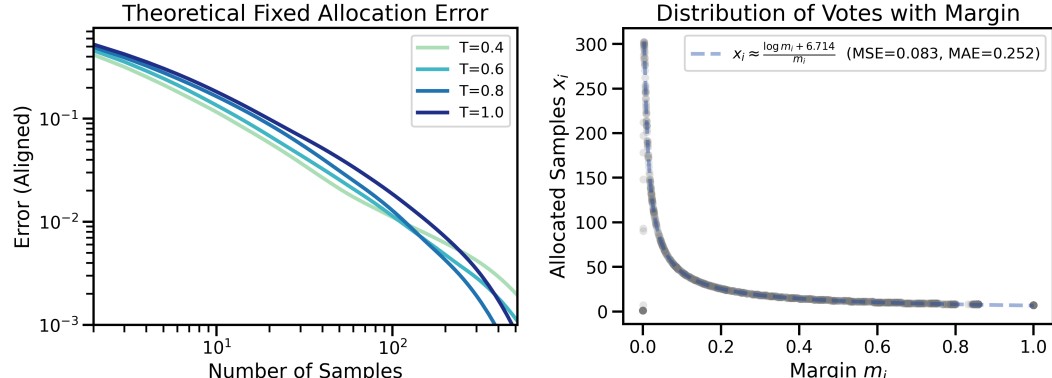

Figure 10: We use the error model $e^{-m_i x_{m_i}}$ using margins extracted from running Llama-3.2-3B on GSM8K. (Left) We observe weak power-law scaling initially that tapers off in the high-sample regime. (Right) Samples closely follow the theoretical distribution from Lagrangian optimality.

*where $\rho$ is a categorical distribution with the same support as $\mu(\,\cdot\,|\,q)$.*

and from Anand Jain et al. (2022), we have

**Theorem 10.** *Let the stopping criteria of PPR-1v1 be $\mathcal{S}_\delta^P$ for any $\delta$. Then $\lim_{\delta \to 0^+} \frac{x(S_\delta^P, q_i)}{\mathsf{LB}(\delta, q_i)} = 1$*

First, at best, we can have exponential error scaling as any specific question has exponential error scaling. Let $q_i = \mathrm{argmin}_{q \in \mathcal{D}} \sup_{\rho:\arg\max(\rho) \neq \arg\max \mu(\cdot|q)} \frac{1}{\mathsf{KL}(\mu(\,\cdot\,|\,q),\rho)}$ and $\rho_i$ be the corresponding distribution. PPR-1v1 achieves the asymptotically optimal exponential scaling as

$$\lim_{\delta \to 0^+} \frac{\mathsf{KL}(\mu(\,\cdot\,|\,q_i),\rho_i)}{\ln(\frac{1}{2.4\delta})}\mathbb{E}_{q_j \sim \mathsf{Unif}(\mathcal{D})}\left[x(\mathcal{S}_\delta^P, q_j)\right] \leq \lim_{\delta \to 0^+} \mathbb{E}_{q_j \sim \mathsf{Unif}(\mathcal{D})}\left[\frac{x(\mathcal{S}_\delta^P, q_j)}{\mathsf{LB}(\delta, q_j)}\right] = 1$$

# D ADDITIONAL RESULTS ON MARGIN CORRELATION

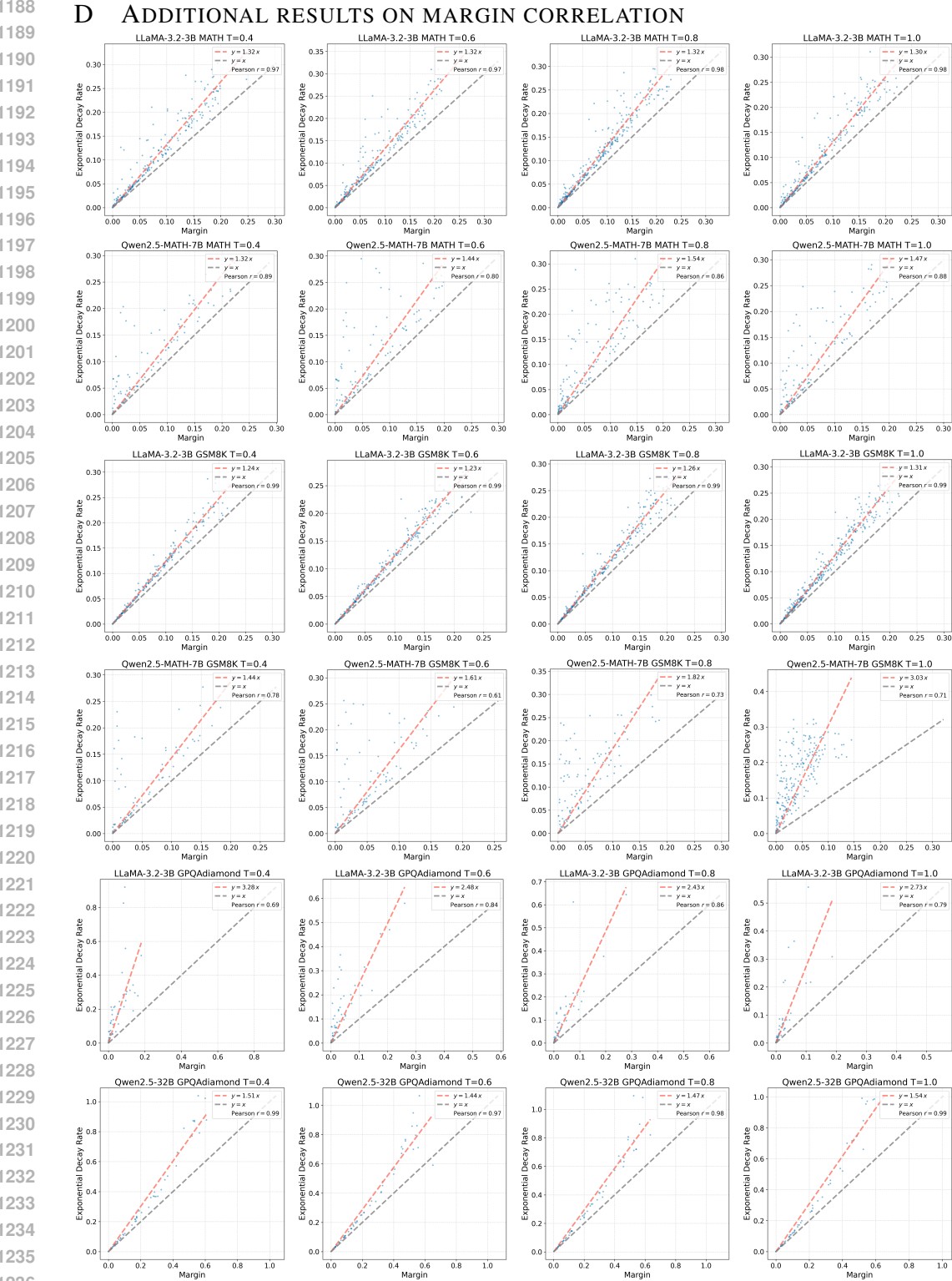

Figure 11: We compare the empirical decay rate with margin, and observe high Pearson correlation. This substantiates our theoretical error model in Section 3

# E    ADDITIONAL RESULTS ON DATASET SCALING LAWS

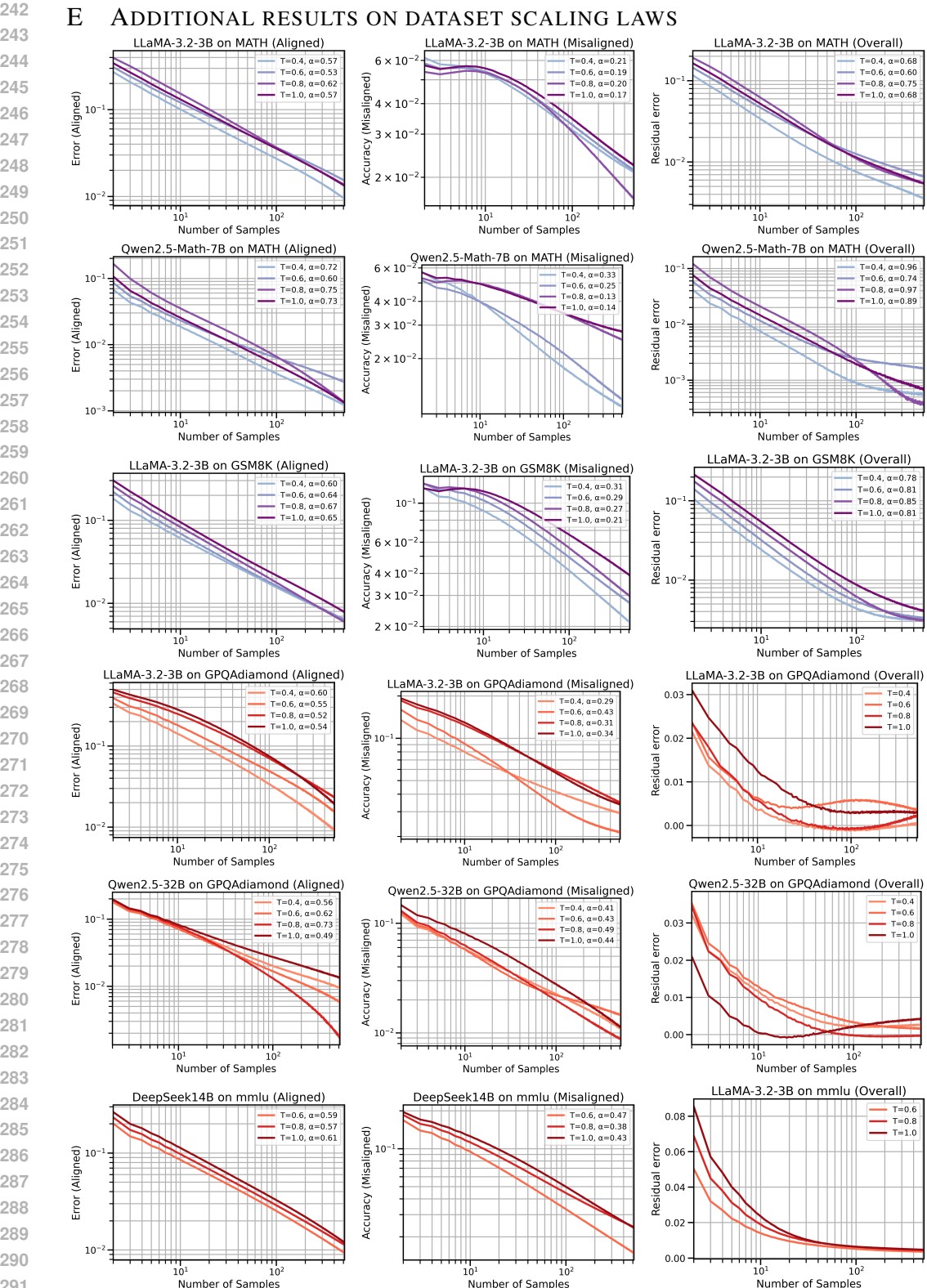

Figure 12: We observe strong power law scaling on aligned questions and weaker power law scaling on misaligned questions. For free-response benchmarks (blue), power-law scaling is prominent up to 100 samples. For multiple-choice benchmarks (red), we cannot reliably predict performance.

# F   ADDITIONAL RESULTS ON BLEND−ASC

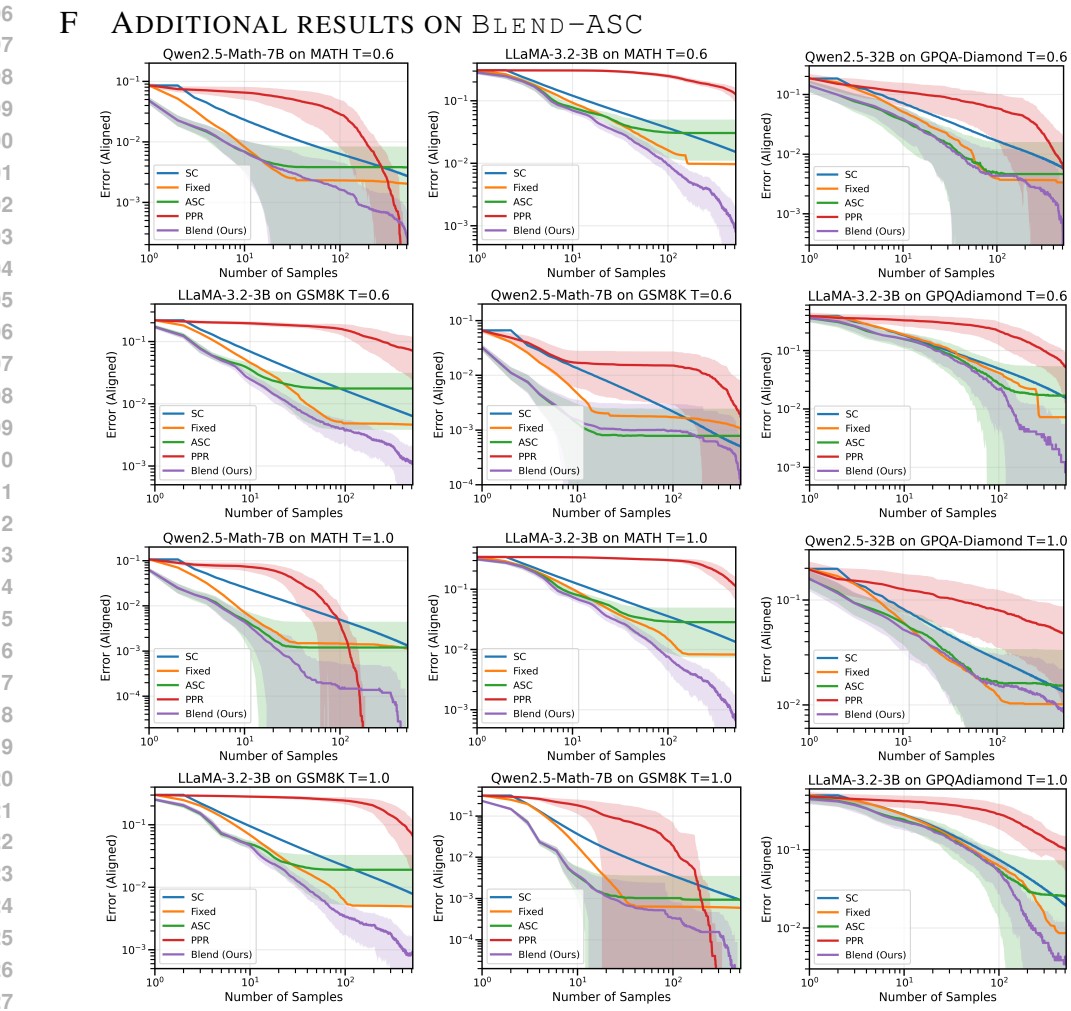

Figure 13: We compare all self-consistency variants on mode estimation across several temperatures.

| SC@n | Algorithm | GSM8K | | MATH | | GPQA-Diamond | | Average Improvement |
|---|---|---|---|---|---|---|---|---|
| | | Llama-3B | Qwen-Math | Llama-3B | Qwen-Math | Llama-3B | Qwen-32B | |
| 64 | Fixed-Allocation | 23 | 13 | 28 | 11 | 56 | 30 | 2.39 × |
| | Adaptive SC | 24 | 7 | 31 | 10 | 45 | 21 | 2.78× |
| | **Blend-ASC (Ours)** | 13 | 6 | 19 | 9 | 38 | 18 | **3.73×** |
| 128 | Fixed-Allocation | 32 | 75 | 43 | 14 | 81 | 42 | 2.68× |
| | Adaptive SC | 128 | 10 | 77 | 14 | 67 | 33 | 2.33× |
| | **Blend-ASC (Ours)** | 19 | 9 | 30 | 11 | 57 | 25 | **5.09×** |

(a) Sample efficiency at temperature 0.6.

| SC@n | Algorithm | GSM8K | | MATH | | GPQA-Diamond | | Average Improvement |
|---|---|---|---|---|---|---|---|---|
| | | Llama-3B | Qwen-Math | Llama-3B | Qwen-Math | Llama-3B | Qwen-32B | |
| 64 | Fixed-Allocation | 15 | 9 | 18 | 10 | 27 | 15 | 4.66× |
| | Adaptive SC | 13 | 6 | 16 | 7 | 33 | 13 | 5.93× |
| | **Blend-ASC (Ours)** | 11 | 6 | 14 | 7 | 26 | 8 | **6.78×** |
| 128 | Fixed-Allocation | 34 | 14 | 45 | 19 | 77 | 37 | 4.60× |
| | Adaptive SC | 128 | 11 | 77 | 13 | 102 | 29 | 4.97× |
| | **Blend-ASC (Ours)** | 22 | 9 | 31 | 12 | 80 | 25 | **6.92×** |

(b) Sample efficiency at temperature 0.8.

| SC@n | Algorithm | GSM8K | | MATH | | GPQA-Diamond | | Average Improvement |
|---|---|---|---|---|---|---|---|---|
| | | Llama-3B | Qwen-Math | Llama-3B | Qwen-Math | Llama-3B | Qwen-32B | |
| 64 | Fixed-Allocation | 23 | 17 | 25 | 11 | 53 | 23 | 2.53× |
| | Adaptive SC | 21 | 10 | 31 | 9 | 49 | 30 | 2.56× |
| | **Blend-ASC (Ours)** | 16 | 9 | 20 | 8 | 49 | 21 | **3.12×** |
| 128 | Fixed-Allocation | 36 | 21 | 41 | 15 | 82 | 37 | 3.31× |
| | Adaptive SC | 128 | 13 | 68 | 13 | 87 | 43 | 2.18× |
| | **Blend-ASC (Ours)** | 25 | 12 | 29 | 10 | 90 | 26 | **4.00×** |

(c) Sample efficiency at temperature 1.0.

Table 5: Comparison of sample efficiency across adaptive methods under different temperatures.

