# OpenReview forum: "Multi-path reasoning on a budget: towards theoretically optimal hyperparameter-free adaptive self-consistency"
_ICLR.cc/2026/Conference — Submitted to ICLR 2026_

### Official Review · Reviewer_gZMe · 2025-10-26

**Soundness:** 3
**Presentation:** 3
**Contribution:** 3
**Rating:** 4
**Confidence:** 1

**Summary:**

The paper propose a theorical provement of efficient adaptive sc together with the experiments prove the theory.

## A bad news is I am not familar with theory provement, I will only focus on the experiments part. Please reduce the weight of my rating, since it is a theory-based research paper.

**Strengths:**

The paper give  a

**Weaknesses:**

Could you briefly clarify the current experimental setup? It seems to be quite confusing for the readers.It might also be helpful to: Compare how 'efficiency' is defined and Compare the final performance under the same sampling conditions (or 'given the same sampling budget').

Could you provide a simple summary in layman's terms? It should explain what signals are used to assist ASC？

Does this require additional time/source to caculate whether to stop?

**Questions:**

See weakness.

---

> ### Author Response · Authors · 2025-11-22
>
> We thank the reviewer gZMe for the transparency of their review and acknowledging that they only focus on the experimental part of our work. We provide below a point-by-point clarification to address their specific questions and try to convey the meaning behind our theoretical contributions that led to Blend-ASC.
>
> ---
> **[Q1] Experimental setup?**
> > 1. Could you briefly clarify the current experimental setup? It seems to be quite confusing for the readers.It might also be helpful to: Compare how 'efficiency' is defined and Compare the final performance under the same sampling conditions (or 'given the same sampling budget').
> ---
> We thank the reviewer for this question. Our experimental setup is as follows. We report in Table 1 the results of running an adaptive SC method and our method, Blend-ASC, on several common LLM evaluation benchmarks. The leftmost column indicates how many reasoning paths an LLM can generate for a given question (referred to as budget). The rightmost column tells us how many CoT generations were saved while reaching the same performance as vanilla SC (x6 with a budget of 128 that we did 128/6 CoT generations, 6 times less). The numbers in the middle give the exact number of calls. Overall, this table shows how many CoT generations one can save per question to get the same accuracy as vanilla SC. Our approach is based on a theoretically proved statement that claims a very fast error decay for when we generate more and more CoT paths.
>
> ---
> **[Q2] Simple summary of the approach?**
> > 2. Could you provide a simple summary in layman's terms? It should explain what signals are used to assist ASC？
> ---
>
> We include an illustrative scheme on page 8 to clarify the inner workings of our method. In layman's terms, self-consistency (SC) improves LLM's performance on a task by asking it to generate several independent responses and then SC selects the most common answer. Despite its simplicity, SC performs well and has become a popular inference scaling approach. When SC is traditionally run over several questions (a dataset), each question generates the same number of responses. This is often wasteful because LLMs can immediately answer easy questions like 1+1=_ every time. However, for difficult questions, LLMs can generate many different answers. In these cases, we should use more generations to find the answer that appears the most often.
>
> Adaptive Self-Consistency methods determine the number of generations during inference. It generates responses for questions and sees if the model is confident (responses are concentrated on one answer) or unsure (responses are spread evenly across several candidate answers). It generates more responses for questions where the model is unsure and iteratively repeats this process. Our method, Blend-ASC, uses two different confidence "scores" to identify the questions where the model is least sure. In addition, our method uses no hyperparameters while other methods require 2.
>
> We would like to summarize the theoretical portion in simple terms, where we make two contributions: 1. We show that ASC is fundamentally the statistical problem of "mode estimation" and comprehensively explore methods to target this problem 2. We derive test-time scaling laws for SC and its variants.
>
> The first point is important as there are many works on SC, but few explore it from first principles: SC predicts the most common answer, or the mode. Our theoretical perspectives give readers an understanding of how SC fundamentally behaves and its strengths and failures. The second point is important as test-time scaling laws are traditionally difficult to identify, but we mathematically delineate SC's behavior and confirm this empirically. Both with theory and experiments, we discover variants that are much more efficient.
>
> ---
> **[Q3] Additional overhead?**
> > 3. Does this require additional time/source to caculate whether to stop?
>
> The additional overhead is negligible. For example, it takes an additional ~1.8 seconds to run Blend-ASC on GPQA Diamond with a budget of 128 samples. From OpenRouter, the throughput of GPT-5 is 37.12 tokens per second. A single chain-of-thought can easily be over 37.12 \* 1.8 < 70 tokens long, exceeding the total time of Blend-ASC, and the GPQA Diamond task requires 25000 chain-of-thought responses.
>
> We thank the reviewer for their valuable feedback, which helps us improve the clarity of our work. We hope to have adequately answered the reviewer's concerns and remain open to further discussion in case some issues remain unaddressed.

---

> > ### Author Response · Authors · 2025-11-27
> > **Looking forward for Reviewer's reply**
> >
> > We thank the reviewer gZMe for their interesting questions.
> >
> > We hope that our comments can address their concerns and highlight our contributions.
> >
> > We gently remind them that the reviewer-author discussion period is ending soon. After that, we may not have the opportunity to respond to their comments.
> >
> > We thank them again for their valuable review and time, and would be happy to answer any further questions.

---

### Official Review · Reviewer_GJkR · 2025-10-31

**Soundness:** 2
**Presentation:** 3
**Contribution:** 2
**Rating:** 4
**Confidence:** 4

**Summary:**

This paper presents a comprehensive theoretical and empirical study of Self-Consistency (SC), a popular test-time inference strategy for reasoning in large language models (LLMs). The authors reinterpret SC as a form of mode estimation and majority voting, deriving tight error bounds and scaling laws that explain its performance across datasets. Building on these insights, they propose Blend-ASC, a novel adaptive and hyperparameter-free variant that dynamically allocates inference samples to questions under a fixed compute budget. Blend-ASC combines the theoretical optimality of PPR-1v1 with the practical efficiency of Adaptive SC, achieving superior sample efficiency—up to 6.8× fewer samples than vanilla SC—across various LLMs (LLaMA-3.2, Qwen2.5) and benchmarks (GSM8K, MATH, MMLU, GPQA-Diamond). Theoretical analysis establishes power-law scaling for dataset-level performance and exponential convergence for aligned questions, both supported by extensive empirical validation.

**Strengths:**

1, Builds a principled bridge between SC and classical statistical learning theory.

2, Blend-ASC is parameter-free and easy to implement.

3, The derivation of the asymptotic bound is mathematically elegant.

4, Theoretical, synthetic, and real-data analyses align convincingly.

**Weaknesses:**

1, More related works should be discussed. e.g. https://aclanthology.org/2024.findings-emnlp.135.pdf, https://arxiv.org/abs/2401.02009, https://arxiv.org/abs/2308.00436. For example, at the same cost, does the proposed method perform better than mirror-consistency, self-contrast & self-check?

2,  The dataset-level theoretical analysis relies on idealized margin distributions, which may not perfectly capture real model behavior.

3,  Real-world runtime or energy cost comparisons would enhance practical relevance.

4, While Blend-ASC is efficient, the paper could better explain its qualitative decision process and failure modes under extreme conditions.

**Questions:**

1, What's the performance comparison between the consistency-based methods and other inference-time methods? e.g. multi-agent systems or other prompting methods like step-back https://arxiv.org/abs/2310.06117. Or let me ask in another way, why should we keep optimizing consistency-based methods, given all other prompting strategies?

2, The method is mainly a prompting engineering work. Can the llm be trained to be better at self-consistency? Which i mean is, the model can be trained to generate with a voting strategy.

---

> ### Author Response · Authors · 2025-11-22
> **Answer Part 1**
>
> We thank the reviewer GJkR for highlighting that **our approach is parameter-free, easy to implement with principled, elegant theory and convincing experiments**. We provide below a point-by-point clarification to address their specific questions.
>
> ---
> **[W1] Related work & [Q1] Why studying consistency-based methods?**
> > 1. More related works + why should we keep optimizing consistency-based methods, given all other prompting strategies?
> ---
>
> We thank the reviewer for the suggested references, for which we added a discussion in the Related Work paragraph of the revised version of our paper.
>
> Below, we provide several perspectives on the importance of self-consistency for the research community.
>
> 1. Self-consistency and related notions have been the subject of study in recent papers at top-tier ML venues (ICLR/NeurIPS/ICML), showing the growing interest of the community for this method (eg, [1-5]). This hints at the fact that there is no definite best test-time scaling strategy as of now and all of them remain worth studying. As for the comparison between them, a recent paper reported that self-consistency can be more efficient than the multi-agent debate approach [6].
>
> 2. We also note that the original idea behind self-consistency [7] was not to improve the LLMs' performance (which was secondary, as shown two years later in [8]), but rather to increase the reliability of LLMs, that is, to output consistent answers to similar prompts. As argued in a recent position paper on the reliability of LLMs [9], self-consistency is among the few approaches that allow for achieving this. Reliability of LLMs remains critical, as seen with discussions on LLM determinism [10]. Self-consistency essentially provides a weak form of determinism: increasing computation converges to one deterministic answer. This particular property of self-consistency can also be successfully used for preference optimization [11], where enforcing self-consistent behaviour was shown to be as efficient as preference optimization with an external reward model.
>
> 3. Overall, we believe that the strengths of other methods do not take away from self-consistency. For all methods that do not use verifiers, self-consistency can be applied to their outputs. For example, we can generate several outputs of the successful "Wait" prompting method [12] and apply self-consistency to their outputs to improve performance. The same reliability benefits of self-consistency could be applied to multi-agent settings: if a multi-agent debate generates many different answers across different random seeds, self-consistency could help.
>
> [1] "A Theoretical Study on Bridging Internal Probability and Self-Consistency for LLM Reasoning", NeurIPS 2025
>
> [2] "Self-Consistency Preference Optimization", ICML 2025
>
> [3] "Think Smarter not Harder: Adaptive Reasoning with Inference Aware Optimization", ICML 2025
>
> [4] "Integrative Decoding: Improving Factuality via Implicit Self-consistency", ICLR 2025
>
> [5] "Balancing Act: Diversity and Consistency in Large Language Model Ensembles", ICLR 2025
>
> [6] "Large language models cannot self-correct reasoning yet", ICLR'24.
>
> [7] "Measuring and Improving Consistency in Pretrained Language Models", ACL'21.
>
> [8] "Self-consistency improves chain of thought reasoning in language models", ICLR'23.
>
> [9] "Consistency in Language Models: Current Landscape, Challenges, and Future Directions",
> ICML Workshop on Reliable and Responsible Foundation Models, 2025.
>
> [10] "Defeating Nondeterminism in LLM Inference", Thinking Machines Lab: Connectionism.
>
> [11] Self-Consistency Preference Optimization, ICML'25.
>
> [12] "s1: Simple test-time scaling", EMNLP 2025.
>
> ---
> **[W2] Dataset level analysis**
> > 2. The dataset-level theoretical analysis relies on idealized margin distributions, which may not perfectly capture real model behavior.
> ---
>
> While we understand the reviewer's concern, we kindly note that, as shown in Figure 4, the margin distributions studied in our theoretical analysis (leftmost plot) are close in shape to those observed in practice on real-world benchmarks across several models (middle plot). In fact, our theoretical assumptions were strongly motivated by the observed empirical behaviour in the first place. This ensures that the theory is faithful to real model behavior. We have modified the plot to put them side by side in order to better reflect this proximity.
>
> We hope that this clarifies the raised concern for the reviewer and will allow them to reconsider their stance in this regard.

---

> ### Author Response · Authors · 2025-11-22
> **Answer Part 2 (Final)**
>
> ---
> **[W3] Real-world runtime**
> > 3. Real-world runtime or energy cost comparisons would enhance practical relevance.
> ---
> Unfortunately, we do not have sufficient funds to provide such a study, as it is very costly for frontier models. Indeed, assuming using a GPT-o1 model with a budget of 100 samplings, such an experiment would cost us around $5$ per question (0$.15$x$100$  input tokens + $0.6$x$800$ output tokens for CoT answers) for a total of $1000$ only for GPQA-Diamond benchmark.
>
> If the reviewer thinks that such an experiment may have a decisive role in making them strongly endorse our paper, we can do such an experiment with a cheaper model during the time that remains before the end of the rebuttal.
>
> ---
> **[W4] Decision process and failure modes**
> > 4. While Blend-ASC is efficient, the paper could better explain its qualitative decision process and failure modes under extreme conditions.
> ---
>
> We thank the reviewer for the suggestion to improve the explanation of the decision process behind Blend-ASC. We address it on page 8 in the revised version of our paper by clarifying how the Blend-ASC approach works. As for the failure modes, Blend-ASC is at least as good as the adaptive approach ASC, which remains a strong baseline. Additionally, Blend-ASC enjoys a favourable exponential decay rate of PPR in the high sample regime.
>
> As for the extreme conditions, we note that external verifier-free methods (including self-consistency) are not able to correct systematic errors in the underlying base model but only to amplify existing signal. If the model is too weak for a given benchmark, it requires a separately trained reward model to fix this.
>
> ---
> **[Q2] Prompt engineering method?**
> > 5. The method is mainly a prompting engineering work. Can the llm be trained to be better at self-consistency? Which i mean is, the model can be trained to generate with a voting strategy.
>
> We respectfully disagree with the reviewer when casting our work as "prompting engineering". In fact, **we do not change the prompt at all**, but instead efficiently allocate compute resources during inference. Additionally, as noted by the reviewer in their strengths, our paper "builds a principled bridge between SC and classical statistical learning theory" when doing so. As such, our work is first and foremost a theoretical study of test-time scaling laws for self-consistency and its adaptive variants, improving over prior works by obtaining a tighter bound for the error decay, deriving it for a dataset-level performance, and validating our findings experimentally. Such scaling laws provide concrete guidance for the expected gains with the increased computational budget that one may expect in practice.
>
> While incorporating self-consistency may present an interesting approach to improving LLMs' pre-training, such a study is out of scope of this work, which concentrates on the test-time scaling rather than pre-training. In this sense, self-consistency is more useful for RLHF, as shown in [1], which used self-consistency for preference optimization. We thank the reviewer for the insightful suggestion, which we included in Section 6 of our work. We hope our answer clarifies our main contributions and the scope of our work to the reviewer.
>
> *References*
>
> [1] Self-Consistency Preference Optimization, ICML'25.
>
> We thank the reviewer for their valuable feedback, which has helped us improve the clarity of our work. We hope to have adequately answered the reviewer's concerns and remain open to further discussion in case some issues remain unaddressed.

---

> > ### Comment · Reviewer_GJkR · 2025-11-23
> >
> > Thank you for your response, which has resolved some of my confusion. I believe there is still room for further improvement and refinement in this work, and I am pleased that you are open to these suggestions. I have ultimately decided to maintain my original rating.

---

> ### Author Response · Authors · 2025-11-24
> **Any Remaining Concerns?**
>
> We are happy to read that our rebuttal correctly addressed some of the reviewer's questions and thank them for their encouraging answer.
>
> Could the reviewer please point us to the remaining questions in their original review that motivated their rating and are not already addressed in our first rebuttal?

---

> > ### Author Response · Authors · 2025-11-27
> > **Looking forward for Reviewer's reply**
> >
> > We thank the reviewer GJkR for their insightful feedback, which we believe helped us improve our submission.
> >
> > We appreciate reading that we correctly addressed some of the reviewer's questions. We would be happy to further address any remaining questions the reviewer might have that are motivating their current rating.
> >
> > We gently remind them that the reviewer-author discussion period is ending soon. After that, we may not have the opportunity to respond to their comments.
> >
> > We thank them again for their valuable review and time, and would be happy to answer any further questions.

---

### Official Review · Reviewer_3NRg · 2025-11-01

**Soundness:** 3
**Presentation:** 3
**Contribution:** 3
**Rating:** 6
**Confidence:** 4

**Summary:**

This paper revisits SC and provides a clean, interpretable bound for SC as a mode estimation method (exponential decay at rate $ (\sqrt{p_1}-\sqrt{p_2})^2$, Theorem 1). In simpler terms, cases where the margin between the top two answers is less take more samples to converge. This “margin” lens also explains dataset‑level power‑law behavior. On the methods side, they take two reasonable ideas: ASC (good early) and PPR‑1v1 (asymptotically optimal late), and blend them into a new method (Blend-ASC) that spends samples where the top-2 are unstable. Empirically, they report ~6.8 times fewer samples than vanilla SC to match accuracy.

**Strengths:**

* Clear and usable bound: exponential decay with an interpretable margin that practitioners can understand; accurately characterizes behaviors across models and datasets (see Fig. 3). This is, in my view, the most impressive aspect of the paper.

* A blend of two sensible policies: They combine two methods with complementary strengths, ASC for speed early and PPR-1v1 for guarantees late, to achieve good practical results.

**Weaknesses:**

* The main theory and many plots emphasize aligned items (the majority answer is correct). The misaligned behavior is weaker, and MC is sometimes non-monotonic (Fig. 5). This is not a weakness per se, but it does make the results somewhat less compelling.

* Practical impact is unclear:
  * How to set budgets, handle latency/cost trade‑offs, partial allocations across streaming datasets, and interaction with caching/batching is unclear.
  * If a team can only perform “SC@N per query” (without cohorting), is there a lightweight variant of Blend-ASC that still achieves a decent fraction of the gains?

**Questions:**

* For real-world evaluation, can you report at least one end-to-end LLM experiment with wall-clock and monetary costs to demonstrate that Blend-ASC retains its advantage?






* Suggestion for (exciting) future work: extend the analysis to “thinking” models by replacing additional samples with additional thinking steps, and test whether longer thinking on harder questions outperforms extra votes under realistic cost/latency constraints.

---

> ### Author Response · Authors · 2025-11-22
> **Answer Part 1**
>
> We thank the reviewer 3NRg for highlighting **the clarity, interpretability, and practical utility of our approach**. This kind of feedback feels very encouraging. We provide below a point-by-point clarification to address their specific questions.
>
> ---
> **[W1] Misaligned setting**
> > 1. The main theory and many plots emphasize aligned items (the majority answer is correct). The misaligned behavior is weaker [...] This is not a weakness per se, but it does make the results somewhat less compelling.
> ---
> We thank the reviewer for this question. We first note that the alignment assumption is a restriction of self-consistency and all verifier-free parallel scaling techniques, rather than an assumption introduced artificially by our work. To put it simply, without an external oracle/feedback, we cannot correct systematic errors in the underlying base model. We can only amplify the existing signal, identifying the mode, and seek to do this more efficiently (which our method does). From Appendix E, we see that for free-response answers, aligned questions dominate the scaling behavior (in blue). This is because when the true answer is not the mode, it often has negligible probability mass. Thus, misaligned questions become an irreducible error.
>
> Second, deriving a tight theoretical guarantee for the misaligned part requires an infeasible combinatorial analysis that depends on the number of possible rankings between the most likely answer from the LLM's perspective and the true answer, as well as the probabilities of each answer in between. We note our theoretical guarantees are asymptotically tight for misaligned questions, and we have a power law decrease in accuracy. Furthermore, predicting downstream scaling for multiple-choice benchmarks is notoriously difficult, as shown in [1].
>
> [1] Why Has Predicting Downstream Capabilities of Frontier AI Models with Scale Remained Elusive? Shaeffer et al., 2025.
>
> ---
> **[W2] Practical impact**
> > 2. Practical impact is unclear: how to set budgets, handle latency/cost trade‑offs
> ---
> We thank the reviewer for this question, which is of high practical importance. For setting budgets, an upper bound can be determined by a user's overall compute budget. The unique benefit of Blend-ASC is that, unlike ASC and ESC, users can observe how question confidences change with time and adjust their latency/cost trade-off accordingly. A user can do this by prematurely ending the process (to save compute and time) when sufficient confidence is reached. For ASC and ESC, the amount sampled is non-deterministic and requires tuning of two hyperparameters, which is expensive and complicated (especially given the setting of LLM inference **scaling**). Since they sample question by question, they cannot prematurely end, as that leaves whole questions without answers. We believe these practical benefits make Blend-ASC much more compelling to use for practitioners.
>
> ---
> **[W3] Caching/Batching**
> > 3. partial allocations [...] and interaction with caching/batching is unclear. If a team can only perform “SC@N per query” (without cohorting), is there a lightweight variant of Blend-ASC that still achieves a decent fraction of the gains?
> ---
> We thank the reviewer for this suggestion. To allow for a batched/low-latency setting, we carefully implemented two variations of Blend-ASC as follows.
>
> First variation uses batched CoT samplings with parallel calls to LLM. The main difference here is that we identify $b$ hardest question rather than 1 as used in the original version of Blend-ASC. The table below (included in the revised version of our paper for the budget of 64 as well) shows that such an approach doesn't affect the performance. As before, each number here represents the sample complexity improvement compared to SC.
>
> | SC@n | Batch Size | T=0.4 | T=0.6 | T=0.8 | T=1.0 |
> |------|------------|-------|-------|-------|-------|
> | 128  | 1  | 8.40 | 7.37 | 6.93 | 6.56 |
> | 128  | 2  | 6.62 | 5.79 | 6.50 | 5.35 |
> | 128  | 4  | 6.72 | 5.95 | 6.34 | 5.43 |
> | 128  | 8  | 7.65 | 7.33 | 5.56 | 5.14 |
> | 128  | 16 | 6.81 | 6.71 | 6.23 | 5.19 |
> | 128  | 32 | 6.44 | 6.11 | 5.54 | 5.15 |
>
> The second variation of Blend-ASC is to group the queries until they reach a desired batch size $b$ and run Blend-ASC on each such batch. We present the results for this approach below (and in the revised manuscript):
>
> | SC@n | #Groups | T=0.4 | T=0.6 | T=0.8 | T=1.0 |
> |------|----------|-------|-------|-------|-------|
> | 128  | 1  | 8.40 | 7.37 | 6.93 | 6.56 |
> | 128  | 2  | 7.84 | 7.16 | 6.53 | 6.24 |
> | 128  | 4  | 7.91 | 7.37 | 6.47 | 6.11 |
> | 128  | 8  | 7.03 | 6.42 | 6.27 | 5.73 |
> | 128  | 16 | 6.54 | 6.17 | 6.01 | 5.43 |
>
> When the number of considered groups increases, ie, when the batch size becomes smaller and we approach an online setting, Blend-ASC still leads to substantial sample complexity improvements.
>
> We believe that this provides strong evidence for the applicability of Blend-ASC in low-latency settings.

---

> ### Author Response · Authors · 2025-11-22
> **Answer Part 2 (Final)**
>
> ---
> **[Q1] Real-world evaluation**
> > 3. For real-world evaluation, can you report at least one end-to-end LLM experiment with wall-clock and monetary costs to demonstrate that Blend-ASC retains its advantage?
> ---
> Unfortunately, we do not have sufficient funds to provide such a study, as it is very costly for frontier models. For a common GPT-o1 model, assuming a budget of 100 samplings, such an experiment would cost us around $5$ per question ($0.15$x$100$  input tokens + $0.6$x$800$ output tokens for CoT answers) to a total of $1000$ only for GPQA-Diamond benchmark.
>
> If the reviewer thinks that such an experiment may have a decisive role in making them strongly endorse our paper, we can do such an experiment with a cheaper model during the time that remains in the discussion.
>
> ---
> **[Q2] Exciting future work**
> > 4. Suggestion for (exciting) future work: extend the analysis to “thinking” models by replacing additional samples with additional thinking steps, and test whether longer thinking on harder questions outperforms extra votes under realistic cost/latency constraints.
> ---
>
> We thank the reviewer for this interesting suggestion, which we will consider for future work. The model's performance would improve as additional thinking steps likely increase the probability of mode alignment. This is complementary to our work, which improves the efficiency for aligned questions, and provides a new axis to scaling. The reviewer's proposed method would improve the model's inherent reasoning capacity, thus the proportion of aligned questions, while our method maximizes the model's performance on these aligned questions, efficiently achieving its capacity.
>
> We note that the reviewer's suggestion aligns with sequential test-time scaling, where we increase compute for one generation. The authors of [1] demonstrate that optimal test-time scaling can be a combination of both sequential scaling and parallel scaling (such as Self-Consistency), again highlighting the proposed method's synergy.
>
> We thank the reviewer for their valuable feedback, which helps us improve the clarity of our work. We hope to have adequately answered the reviewer's concerns and remain open to further discussion in case some issues remain unaddressed.
>
> [1] "Scaling LLM Test-Time Compute Optimally can be More Effective than Scaling Model Parameters", ICLR 2025.

---

> > ### Author Response · Authors · 2025-11-27
> > **Looking forward for Reviewer's reply**
> >
> > We thank the reviewer 3NRg for their encouraging feedback, which we believe helped us improve our submission.
> >
> > We answered all of the reviewers' questions and conducted the additional experiments they requested. We hope that our comments can address their concerns and highlight our contributions.
> >
> > We gently remind them that the reviewer-author discussion period is ending soon. After that, we may not have the opportunity to respond to their comments.
> >
> > We thank them again for their valuable review and time, and would be happy to answer any further questions.

---

### Official Review · Reviewer_z7NJ · 2025-11-01

**Soundness:** 3
**Presentation:** 3
**Contribution:** 2
**Rating:** 4
**Confidence:** 4

**Summary:**

This paper studies test-time scaling via self-consistency (SC) sampling for large language models. The authors formalize SC as a majority-vote estimation problem and introduce a margin quantity defined by the gap between the top-2 answer probabilities. Under an aligned assumption—where the correct answer corresponds to the mode—the paper establishes an exponential error decay rate governed by this margin. At the dataset level, the error rate can be expressed as a Laplace transform over the margin distribution, which induces power-law scaling. Building on these insights, the authors propose a compute-allocation strategy (Blend-ASC) that blends an asymptotically optimal but conservative pairwise comparison rule with a more aggressive adaptive allocation heuristic. Experiments based on resampling from limited initial generations suggest that Blend-ASC reduces the number of samples required to match vanilla SC performance on several benchmarks.

**Strengths:**

### Originality
The paper provides a clean and unified theoretical framing of SC through a classical margin-based lens. Relating dataset-level behavior to the margin distribution helps consolidate several empirical scaling observations.

### Quality
The theoretical arguments are well-organized, and assumptions are clearly stated. The proposed algorithm is reasonable from a systems perspective and demonstrates consistent improvements under the authors’ experimental protocol.

### Clarity
The paper is well written, with helpful intuition behind proofs and algorithmic components. The distinction between aligned and misaligned questions is explicitly surfaced, which avoids overstating the theoretical coverage.

**Weaknesses:**

**Practical utility of the theory**
The exponential decay guarantees require strong alignment assumptions. In settings where the model’s top mode is systematically wrong, margin-based confidence can be misleading, limiting the operational usefulness of Section 4.1.

**Relative novelty and applicability**
Compared to prior adaptive schemes such as ASC and ESC, the proposed approach does not appear to introduce a substantially new conceptual perspective on the core allocation problem; rather, it refines similar margin-based heuristics and inherits some of their limitations.


**Batch inference assumption**
The algorithm fundamentally assumes a global batch of questions and a shared budget. Many realistic deployments are online or have per-query latency constraints. The paper does not discuss feasibility or degradation in such scenarios.

**Questions:**

**Misaligned regimes:**
   When the model’s top mode is incorrect, do margin-based confidence estimates systematically misallocate budget? Are there diagnostics or caps that prevent over-exploration of misleading modes?

**Online/latency settings:**
   How might Blend-ASC be adapted when queries arrive incrementally or must be answered individually under strict latency? Is there a small-batch approximation that preserves most gains?

---

> ### Author Response · Authors · 2025-11-22
> **Answer Part 1**
>
> We thank the reviewer z7NJ for highlighting **the originality and quality** our theoretical contributions to self-consistency. We provide below a point-by-point clarification to address their specific questions.
>
> ----
> **[W1] Practical utility of the theory**
> > 1. Practical utility of the theory; The exponential decay guarantees require strong alignment assumptions. In settings where the model’s top mode is systematically wrong, margin-based confidence can be misleading, limiting the operational usefulness of Section 4.1.
> ---
> We thank the reviewer for this important question. We first note that the alignment assumption is a restriction of self-consistency and all verifier-free parallel scaling techniques, rather than an assumption introduced artificially by our work. To put it simply, without an external oracle/feedback, we cannot correct systematic errors in the underlying base model. We can only amplify the existing signal, identifying the mode, and seek to do this more efficiently (which our method does). From Appendix E, we see that for free-response answers, aligned questions dominate the scaling behavior (in blue). This is because when the true answer is not the mode, it often has negligible probability mass. Thus, misaligned questions become an irreducible error.
>
> Second, deriving a tight theoretical guarantee for the misaligned part requires an infeasible combinatorial analysis that depends on the number of possible rankings between the most likely answer from the LLM's perspective and the true answer, as well as the probabilities of each answer in between. We note our theoretical guarantees are asymptotically tight for misaligned questions, and we have a power law decrease in accuracy. Furthermore, predicting downstream scaling for multiple-choice benchmarks is notoriously difficult, as shown in [1].
>
> We added a remark after Theorem 1 to elaborate on this.
>
> [1] Why Has Predicting Downstream Capabilities of Frontier AI Models with Scale Remained Elusive? Shaeffer et al., 2025.
>
> ----
> **[W2] Relative novelty and applicability**
> >  2. novelty and applicability; Compared to prior adaptive schemes such as ASC and ESC, the proposed approach does not appear to introduce a substantially new conceptual perspective on the core allocation problem; rather, it refines similar margin-based heuristics and inherits some of their limitations.
> ---
> We thank the reviewer for giving us the possibility to clarify this. We note that our work derived Blend-ASC as a principled approach from theoretically proved test-time scaling laws. We carefully derived scaling laws for self-consistency ($O(n^{1/2}$) and identified a stopping condition (PPR) that achieves an exponential decay rate. Kindly note that our work is the first to show that PPR can be used as a stopping criterion in adaptive self-consistency. Our approach to combine it with ASC is novel simply because PPR has never been used in the self-consistency field before. We hope this clarifies the positioning of our method to the reviewer.
>
> Finally, we note that the "limitations of margin-based methods" are not from margin-based methods but SC itself. Yet, within this framework, Blend-ASC is much more applicable than other methods. We don't require any hyperparameter tuning, and our method guarantees a set budget. Following the reviewer's comments, we also demonstrate batch inference support and early stopping capability, which increases the applicability of Blend-ASC in practice (see answer below).

---

> ### Author Response · Authors · 2025-11-22
> **Answer Part 2 (Final)**
>
> ---
> **[W3] Batch inference assumption & [Q2]  Online/latency settings**
> > 3. The algorithm assumes a global batch of questions and a shared budget. How might Blend-ASC be adapted when queries arrive incrementally or must be answered individually under strict latency? Is there a small-batch approximation that preserves most gains? (Weakness X, Question Y)
> ---
> We thank the reviewer for this insightful suggestion. To allow for a batched/low-latency setting, we carefully implemented two variations of Blend-ASC and conducted additional experiments over 3 benchmarks, 6 models, and 4 temperature values (see also PDF at the end of Section 5).
>
> First variation uses batched CoT samplings to leverage the capacity of an LLM to run parallel calls. The main difference here is that we identify $b$ hardest question rather than 1 as used in the original version of Blend-ASC. The table below (included in the revised version of our paper for the budget of 64 as well) shows that such an approach doesn't affect the performance. As before, each number here represents the sample complexity improvement compared to SC.
>
> | SC@n | Batch Size | T=0.4 | T=0.6 | T=0.8 | T=1.0 |
> |------|------------|-------|-------|-------|-------|
> | 128  | 1  | 8.40 | 7.37 | 6.93 | 6.56 |
> | 128  | 2  | 6.62 | 5.79 | 6.50 | 5.35 |
> | 128  | 4  | 6.72 | 5.95 | 6.34 | 5.43 |
> | 128  | 8  | 7.65 | 7.33 | 5.56 | 5.14 |
> | 128  | 16 | 6.81 | 6.71 | 6.23 | 5.19 |
> | 128  | 32 | 6.44 | 6.11 | 5.54 | 5.15 |
>
> The second variation of Blend-ASC is to group queries until they reach a desired batch size $b$ and run Blend-ASC on each such batch. We present the results for this approach below (and in the revised manuscript):
>
> | SC@n | #Groups | T=0.4 | T=0.6 | T=0.8 | T=1.0 |
> |------|----------|-------|-------|-------|-------|
> | 128  | 1  | 8.40 | 7.37 | 6.93 | 6.56 |
> | 128  | 2  | 7.84 | 7.16 | 6.53 | 6.24 |
> | 128  | 4  | 7.91 | 7.37 | 6.47 | 6.11 |
> | 128  | 8  | 7.03 | 6.42 | 6.27 | 5.73 |
> | 128  | 16 | 6.54 | 6.17 | 6.01 | 5.43 |
>
> When the number of considered groups increases, i.e., when the batch size becomes smaller and we approach an online setting, Blend-ASC still leads to substantial sample complexity improvements.
>
> We believe that this provides strong evidence for the applicability of Blend-ASC in low-latency settings.
>
> ---
> **[Q1] Misaligned regimes**
> > 4. Misaligned regimes: When the model’s top mode is incorrect, do margin-based confidence estimates systematically misallocate budget? Are there diagnostics or caps that prevent over-exploration of misleading modes?
> ---
> We thank the reviewer for the insightful question. We note that it is infeasible to determine the "correctness" of the mode without a verifier. So margin-based estimates (and all mode estimation approaches) naturally misallocate budget when the mode is incorrect. However, this is natural to all test-time compute methods. For example, in Best-of-N, questions where the correct answer has a low probability lead to significant misallocated budgets.
>
> As for the second question, we prevent over-exploration using a budget cap of 16 times the current average allocation.
>
> We thank the reviewer for their valuable feedback, which helps us improve the clarity of our work. We hope to have adequately answered the reviewer's concerns and remain open to further discussion in case some issues remain unaddressed.

---

> > ### Author Response · Authors · 2025-11-27
> > **Looking forward for Reviewer's reply**
> >
> > We thank the reviewer z7NJ for their insightful feedback, which we believe helped us improve our submission.
> >
> > We answered all of the reviewers' questions and conducted the additional experiments they requested. We hope that our comments can address their concerns and highlight our contributions.
> >
> > We gently remind them that the reviewer-author discussion period is ending soon. After that, we may not have the opportunity to respond to their comments.
> >
> > We thank them again for their valuable review and time, and would be happy to answer any further questions.

---

### Author Response · Authors · 2025-11-22
**Rebuttal Summary**

Dear Reviewers, Area Chairs, Senior Chairs, and Program Chairs,

We would like to thank all the reviewers for carefully reading our paper and for their insightful comments. We are deeply grateful to them for acknowledging **the novelty and importance of our theoretical contributions** (Reviewers z7NJ, GJkR, 3NRg), which provide **useful bounds for practitioners** (Reviewer 3NRg) validated with **convincing experiments** (Reviewers GJkR).


Below is a brief summary of the review and rebuttal process.

---

**Scope of our contributions**

Before providing individual replies to all the reviewers, we would like to highlight that our paper's main contributions are to theoretically study a challenging and previously understudied topic of test-time scaling laws for self-consistency and its adaptive version in a full dataset setting. In this sense, our proposed method Blend-ASC should be seen as an immediate by-product of these findings, and not as a sole contribution of our work. We would like to encourage the reviewers to take this into account when assessing our paper during the discussion period.

---

**Changes during rebuttal**

We highlight below the changes we made to take into account the reviewers' comments, along with additional experiments following their suggestions. Changes in the PDF are highlighted in red.

1. **Additional experiments** (reviewers z7NJ, 3NRg): new experimental study in batched setting (Table 2 in the main paper) showing that we get the same improvement when we use Blend-ASC over batches smaller than the set budget (either when generating CoT reasoning paths or when running Blend-ASC in a streaming small batch setting). This large-scale experiment is done over **3 benchmarks, 6 models, and 4 different temperatures**. Results are reported at the end of Section 5.

2. **Clarifications** (reviewers z7NJ, 3NRg, GJkR, gZMe): Additional discussions on the importance of self-consistency in the literature (page 1), alignment assumption (Remark after Theorem 1), a diagram explaining how Blend-ASC is calculated (page 8), and new references suggested by reviewers in related work (page 8).
---

We strongly believe that the reviewers' feedback helped us improve the quality of our paper, and we are thankful to them for their suggestions. We are confident that the clarifications, revisions, and additional experiments have strengthened our contributions.

Finally, we would like to thank the Program Chairs, Senior Chairs, and Area Chairs for their time during the reviewing process. We are looking forward to the final decision and hope our paper can be valuable to the ICLR audience.

Best regards,

The Authors

---

### Meta-Review · Area_Chair_Zn5Q · 2025-12-30

**Summary:**

This paper investigates the popular Self-Consistency (SC) approach and presents a comprehensive theoretical and empirical study. In particular, the authors leverage the mode estimation and voting theory and proposed a Blend-ASC method that dynamically allocates samples. Experimental results on benchmarks demonstrate the efficiency of the proposed approach compared to SC and other baselines.

Reviewers agreed that this paper studies an important problem. The integration of ASC and PPR‑1v1 is well motivated, and the proposed Blend-ASC approach is parameter-free and easy to implement. In addition, the paper is well written, which includes helpful intuition behind proofs and algorithmic components.

However, reviewers also raised many concerns regarding the current version of this work, such as novelty compared to recent variants of SC, practical utility of the theory, assumptions, missing related work, dataset-level analysis, runtime analysis, etc.

**Reviewer Concerns:**

The authors have provided detailed responses with some additional results in their rebuttal. Some of the concerns from reviewers, such as experimental setup, related work, and caching/batching results, have been well addressed. Meanwhile, several other concerns regarding novelty, practical utility of theory, and runtime analysis are not sufficiently addressed by the rebuttal.

**Reviewer Scores:**

Initially, this paper received borderline ratings: 4, 4, 4, and 6. Considering some of the concerns have been addressed by the authors' rebuttal, I think at least one of the reviewers would increase their score from 4 to 6. Overall, this paper is still a borderline case.

---

### Decision · Program_Chairs · 2026-01-26

Reject